# From Seeing to Thinking: Decoupling Perception and Reasoning Improves Post-Training of Vision-Language Models

**Juncheng Wu** [1,2]  **Hardy Chen** [2]  **Haoqin Tu** [2]  **Xianfeng Tang**  **Freda Shi** [3,4]  **Hui Liu**  **Hanqing Lu** [1]  **Cihang Xie** [2]  **Yuyin Zhou** [2]

**Project Page:** https://ucsc-vlaa.github.io/VLM-CapCurriculum/

## Abstract

Recent advances in vision-language models (VLMs) emphasize long chain-of-thought reasoning; yet, we find that their performance on visual tasks is primarily limited by a lack of visual perception as opposed to reasoning itself. In this work, we systematically study the interplay between perception and reasoning in VLM post-training by decomposing their capabilities into three separate training stages: visual perception, visual reasoning, and textual reasoning, incorporating specialized training data. We demonstrate that visual perception (a) requires targeted optimization with specialized data; (b) serves as a fundamental scaffold that should be solidified through staged training before refining visual reasoning; and (c) is more effectively learned via RL than caption-based SFT. Our experiments across multiple VLMs demonstrate that staged training consistently improves both visual perception and reasoning performance over merged training. Notably, models trained with our approach achieve 1.5% higher reasoning accuracy with 20.8% shorter reasoning traces, suggesting that superior perception reduces the need for excessive reasoning. Furthermore, we show that this capability-based staging represents a new curriculum dimension orthogonal to traditional difficulty-based curricula, and combining both yields further additive gains. Our staged-training models achieve superior performance among open-weight VLMs, establishing advanced results on several visual math and perception (*e.g.*, +5.2% on We-Math and +3.7% on RealWorldQA) tasks compared with the base counterpart.

[1]Amazon [2]UC Santa Cruz [3]University of Waterloo [4]Vector Institute, Canada CIFAR AI Chair. Correspondence to: Juncheng Wu <jwu418@ucsc.edu>.

*Proceedings of the 43rd International Conference on Machine Learning*, Seoul, South Korea. PMLR 306, 2026. Copyright 2026 by the author(s).

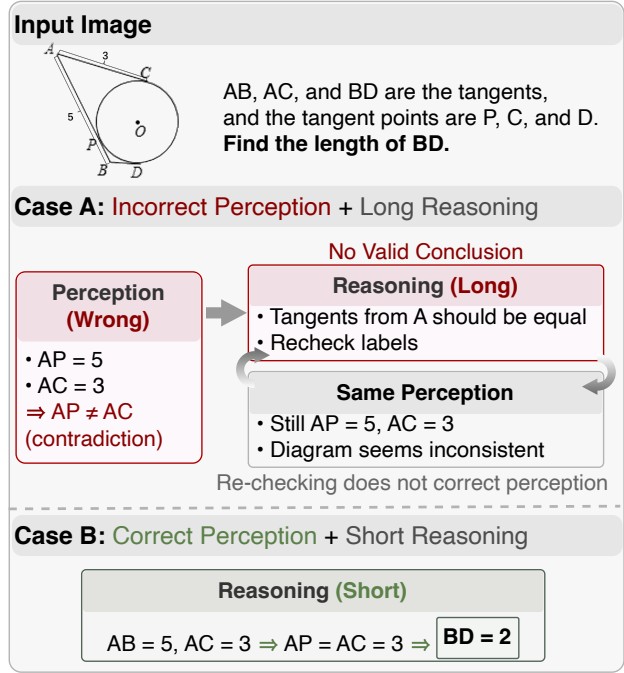

**Input Image**

AB, AC, and BD are the tangents, and the tangent points are P, C, and D. **Find the length of BD.**

**Case A: Incorrect Perception** + Long Reasoning

No Valid Conclusion

**Perception (Wrong)**
- AP = 5
- AC = 3
⇒ AP ≠ AC (contradiction)

**Reasoning (Long)**
- Tangents from A should be equal
- Recheck labels

**Same Perception**
- Still AP = 5, AC = 3
- Diagram seems inconsistent

Re-checking does not correct perception

**Case B: Correct Perception** + Short Reasoning

**Reasoning (Short)**

AB = 5, AC = 3 ⇒ AP = AC = 3 ⇒ **BD = 2**

*Figure 1.* **Longer thinking can not fix incorrect perception.** Re-checking the image during the reasoning leads to the same perception error.

## 1. Introduction

Vision-Language Models (VLMs) have achieved remarkable progress in a wide range of multimodal tasks, including visual question answering (Yue et al., 2024; Huang et al., 2025; Wu et al., 2025), diagram understanding (Hou et al., 2024; Hong et al., 2024), and visual mathematical reasoning (Liu et al., 2023; Wang et al., 2024b; Xu et al., 2025). Recent advances are largely driven by post-training techniques that emphasize long chain-of-thought reasoning via reinforcement learning (RL), enabling models to reason longer for better results (Peng et al., 2025a; Chen et al., 2025; Zhan et al., 2025b; Shen et al., 2025).

However, in many visual reasoning tasks, performance is not primarily limited by reasoning capability but by *visual*

*perception* — *e.g.*, visual mathematics (Lindström & Abraham, 2022; Zhuang et al., 2025), geometry problems (Lu et al., 2023), and diagram-based reasoning (Mathew et al., 2021b). We find that failures in VLM reasoning often stem from the very first visual perception step: once an error is introduced, subsequent reasoning rarely corrects it but instead compounds the mistake based on incorrect perceptual assumptions (see Case A in Figure 1). In contrast, when visual perception is correct, the reasoning becomes concise and converges quickly to the correct answer (Case B). To validate this, we present an analysis of 3 visual math datasets by using the Claude-Haiku-4.5 (Anthropic, 2024) to detect the perception errors in the VLM reasoning process: among all incorrectly sampled answers from Qwen3-VL-8B (Bai et al., 2025a), **86.9%** are due to the visual perception error as described. Both qualitative and quantitative observations, complementing previous works (Ogezi & Shi, 2025; Zhu et al., 2026; Liu et al., 2025), highlight a key limitation of current post-training practices: *longer reasoning does not compensate for incorrect perception.*

We hypothesize that the failure mode may result from flawed post-training paradigms, which emphasize visual reasoning training much more than visual perception in recent studies. We argue that *visual perception should be treated as an independent and fundamental capability in VLMs and trained separately.* To validate our hypothesis, we conduct comprehensive investigations by decoupling VLM capabilities into three stages: visual perception, textual reasoning, and visual reasoning. We propose a staged post-training framework in which each capability is progressively refined using dedicated datasets. In the visual perception stage, we explore the transition from caption based supervised fine-tuning (SFT) to reinforcement learning with verifiable rewards (RLVR). To facilitate this, we construct a scalable data pipeline that transforms standard image-caption datasets (Onoe et al., 2024) into structured, perception-focused training data, allowing the model to close the gap between raw visual input and textual alignment using fully open resources.

Our experimental findings highlight three key factors that are essential for effectively enhancing visual perception in VLMs: (a) **Dedicated data**, similar to textual and visual reasoning, visual perception is not a "solved" pre-training byproduct but requires further targeted optimization with specialized data. On the WeMath benchmark (Qiao et al., 2025), incorporating the visual perception stage in post-training yields a 7.43-point accuracy gain over the Qwen2.5-VL-7B (Bai et al., 2025b) base model and also raises Qwen3-VL-8B performance from 50.9% to 56.1% (Section 4.2); (b) **Staged training**: the staged training paradigm outperforms the common one-stage training setting in which all data for different capabilities are merged and shuffled during post-training. Our staged-trained Qwen3-VL-8B achieves a 1.46-point increase in math reasoning accuracy while producing

20.8% shorter reasoning traces (Section 4.3.1) compared to the one-stage training. Moreover, the order of stage optimization is critical, as visual perception serves as the fundamental scaffold that should be solidified before refining visual reasoning. Disrupting this order reduces the average visual math performance of Qwen2.5-VL-7B from 42.3% to 37.7% (Section 4.3.2); and (c) **RLVR-based visual perception learning**, RLVR provides a significantly more effective training signal for visual perception than caption-based SFT. While SFT can inadvertently degrade performance by imposing token-level, off-policy supervision from data that may be of lower quality than the pre-training corpus, RL keeps the model on-policy, resulting in better alignment. Substituting SFT for RL in visual perception training leads to drops of 8.1% and 1.6% in accuracy for the Qwen2.5-VL-7B and Qwen3-VL-8B models, respectively, on the WeMath benchmark (Section 4.4).

Beyond these empirical findings, our work introduces a conceptual contribution: staged training by capability type can be viewed as *capability-dimension curriculum learning*, a framework orthogonal to traditional difficulty-based curricula. We demonstrate that these two curriculum dimensions are complementary—combining capability-based staging with difficulty-based ordering yields a 4.43% improvement over merged training, surpassing either dimension alone (Section 4.5).

Overall, our staged-training Qwen3-VL-8B attains strong performance on both visual math reasoning (75.9% on Math-Vista and 56.1% on WeMath) and visual perception (74.5% on RealWorldQA) benchmarks (Table 1). Compared to OneThinker-8B, our model improves accuracy by 1.5% on WeMath and 3.0% on RealWorldQA. These findings indicate that integrating our visual perception data with staged-training paradigm yields more advanced reasoning capabilities in VLMs.

**Conflict of Interest Disclosure.** The authors declare no financial conflicts of interest related to this work. Although several authors are affiliated with industry, this study does not evaluate, promote, or rely on any proprietary product or model developed by their employers; all models, datasets, and tools used in our experiments are publicly available.

## 2. Related Work

### 2.1. Reasoning Vision-Language Models

Recent work increasingly targets visual reasoning in VLMs. A common SFT-based direction is to distill structured reasoning traces into the model (Xu et al., 2024; Zhang et al., 2024b; Thawakar et al., 2025; Shao et al., 2024a; Li et al., 2025). In parallel, as DeepSeek-R1 (Guo et al., 2025) gains success in textual reasoning by using Reinforcement Learning with Verifiable Rewards (RLVR) (Shao et al., 2024b),

this paradigm has been adapted to multimodal reasoning to encourage exploration and self-correction (Yang et al., 2025b; Deng et al., 2025c; Peng et al., 2025b; Feng et al., 2025a). Typical vision-related tasks include general visual question answering (VQA) (Marino et al., 2019; Schwenk et al., 2022a; Hudson & Manning, 2019), chart and info-graphic understanding (Masry et al., 2022; Mathew et al., 2021a). Models trained on such tasks with RLVR are enabled to reason over multimodal inputs for higher accuracy. Our approach falls into the same category that leveraging the RLVR approach for tuning a competent reasoning VLM.

### 2.2. Post-training Paradigms For Reasoning VLMs

Post-training for reasoning VLMs typically follows either **merged** training or **curriculum** training. In merged training, diverse supervision signals are merged and optimized together in a single phase. For SFT-based training, LLaVA-CoT exemplifies this by integrating multiple VQA sources with structured reasoning annotations in one training recipe (Xu et al., 2024). For RL-based training, VLAA-Thinker proposes Mixed Reward which blends grounding and reasoning rewards into a single-stage RL training (Chen et al., 2025). Joint training is simple by design but lacks finer-grained considerations on the order of training data. Curriculum learning fills the gap by training models on data with increasing difficulty, manifesting its effectiveness in works like Curr-ReFT (Deng et al., 2025a) and PC-GRPO (Jeddi et al., 2025), which boost performance on both reasoning and perception tasks. Complementary to these training paradigms, recent diagnostic studies have specifically identified visual perception as a key bottleneck. VisOnlyQA (Kamoi et al., 2024) reveals that models struggle with basic geometric understanding through vision-only questions, and NoReGeo (Abdullaeva et al., 2026) isolates perception failures from reasoning by constructing non-reasoning geometry benchmarks. While these works focus on diagnosis, our work addresses the identified gap through a training methodology: instead of sorting data by difficulty, we propose a capability-based curriculum that decouples perception from reasoning and finds that capabilities should be learned following certain orders.

## 3. Staged Post-training Pipeline

### 3.1. Data Synthesis and Curation

We construct three disjoint datasets corresponding to visual perception, textual reasoning, and visual reasoning, respectively. All datasets are synthesized or curated from fully open-source resources.

### 3.1.1. PERCEPTION DATA SYNTHESIS

The objective of the visual perception stage is to improve a model's ability to accurately recognize fine-grained visual details and relative spatial relations without requiring multi-step reasoning.

**Question-Answer Generation from Captions.** We firstly collect image-caption pairs from the DOCCI dataset (Onoe et al., 2024), which contain 15K images paired with fine-grained captions. As shown in Figure 2(a), for each image-caption pair $(I, C)$, we prompt an LLM $f_{\text{gen}}$ (in this work, Qwen2.5-72B) to generate a set of perception-focused question-answer pairs:

$$(Q, A) = f_{\text{gen}}(C) \tag{1}$$

where each question $Q$ emphasizes visual details or spatial relations that are explicitly grounded in the image. The generated answer $A$ serves as the ground truth. The prompt we used is provided in Appendix Figure 7.

**Perception Difficulty Filtering.** To isolate samples that specifically reflect perception deficiencies, we introduce a perception-sensitive filtering criterion as illustrated in Figure 2(b). Let $f_\theta$ denote the base VLM. For each generated question $Q$, we evaluate two inference pathways:

$$\hat{A}_{\text{img}} = f_\theta(I, Q), \quad \hat{A}_{\text{cap}} = f_\theta(C, Q). \tag{2}$$

Where $\hat{A}_{\text{img}}$ refers to the answer to $Q$ by $f_\theta$, with only image $I$ provided, and $\hat{A}_{\text{cap}}$ is the answer generated based on the paired caption. We retain a sample $(I, Q, A)$ if and only if:

$$\mathbb{I}[\hat{A}_{\text{img}} \neq A] \wedge \mathbb{I}[\hat{A}_{\text{cap}} = A], \tag{3}$$

where $\mathbb{I}[\cdot]$ is the indicator function. This condition ensures that the information required to answer $Q$ is present in the caption $C$, while the model fails when relying on its own visual perception from $I$.

To further improve robustness, we apply this filtering using two models, $f_\theta^{(1)} = $ Qwen2.5-VL-7B and $f_\theta^{(2)} = $ Qwen2.5-VL-32B. The resulting dataset $\mathcal{D}_{\text{perc}}$ contains samples that are challenging due to insufficient visual perception rather than reasoning ability. Detailed visual perception data examples are provided in Appendix A.3.

### 3.1.2. REASONING DATA CURATION

For textual reasoning, we use the open-source ORZ-Math-13k dataset (Hu et al., 2025), which consists of challenging math reasoning problems that require multi-step logical inference without visual inputs. The resulting textual reasoning dataset is denoted as $\mathcal{D}_{\text{text}}$.

For visual reasoning, we follow prior work in constructing challenging multimodal reasoning datasets (Chen et al.,

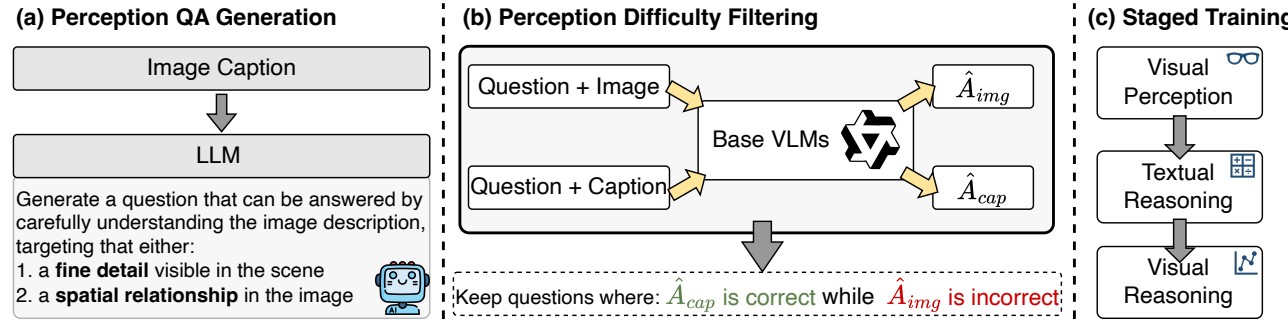

*Figure 2.* **Improving VLM Post-training with Visual Perception Data Synthesis and Staged Training**: (a) Generating image-content based QA pairs by feeding captions to an LLM and labeling answers with a strong VLM; (b) Perception difficulty filtering, which removes samples that can be answered by the base VLMs based on caption; (c) Staged training by different capabilities *from seeing to thinking*.

2025; Xu et al., 2025). We collect samples from multiple open-source sources, including CLEVR-Math (Lindström & Abraham, 2022), GeoQA170K (Gao et al., 2023), Math PUMA (Zhuang et al., 2025), DocVQA (Mathew et al., 2021b), and ArxivQA (Li et al., 2024). We retain samples that require both accurate perception and multi-step reasoning, forming the dataset $\mathcal{D}_{\text{vis}}$.

### 3.2. Training Strategies

#### 3.2.1. STAGED TRAINING

We adopt Group Relative Policy Optimization (GRPO) (Shao et al., 2024b) to enhance the model's reasoning ability without relying on a separate value model. For each input $x$, a group of $G$ responses $\{y_i\}_{i=1}^{G}$ is sampled from the old policy $\pi_{\theta_{\text{old}}}$, and each response is assigned a composite reward $R(x, y_i) = r_{\text{acc}}(x, y_i) + r_{\text{format}}(x, y_i)$. The group-relative advantage is computed by standardizing rewards within each group as:

$$A_i = \frac{R(x, y_i) - \mu_R}{\sigma_R + \epsilon}, \tag{4}$$

where $\mu_R$ and $\sigma_R$ denote the group mean and standard deviation. The policy is then optimized to maximum clipped objective with KL regularization:

$$
\begin{aligned}
\mathcal{J}_{\text{GRPO}}(\theta) =& \\
\mathbb{E}_{x,y} & \left[ \frac{1}{G} \sum_{i=1}^{G} \min\big(\rho_i A_i, \ \text{clip}(\rho_i, 1-\epsilon, 1+\epsilon) A_i\big) \right] \\
& - \beta \, \text{KL}(\pi_\theta \| \pi_{\text{ref}}),
\end{aligned}
\tag{5}
$$

where $\rho_i = \pi_\theta(y_i|x)/\pi_{\theta_{\text{old}}}(y_i|x)$ and $\pi_{\text{ref}}$ is the reference policy from supervised fine-tuning.

In staged training, we optimize the model sequentially over three stages. Each stage is trained for the same number of epochs using identical hyperparameters. The training order

is denoted as:

$$\mathcal{D}_{\text{perc}} \to \mathcal{D}_{\text{text}} \to \mathcal{D}_{\text{vis}}. \tag{6}$$

#### 3.2.2. MERGED TRAINING

For comparison, we construct a merged training baseline by combining all datasets: $\mathcal{D}_{\text{merged}} = \mathcal{D}_{\text{perc}} \cup \mathcal{D}_{\text{text}} \cup \mathcal{D}_{\text{vis}}$. The model is trained on $\mathcal{D}_{\text{merged}}$ with identical hyperparameters and the same total number of steps, reflecting common post-training practices in which perception and reasoning supervision are jointly optimized.

## 4. Experimental Analysis

### 4.1. Experimental Setup

**Models.** We conduct experiments on two VLM backbones Qwen3-VL-8B-Instruct (Bai et al., 2025a) and Qwen2.5-VL-7B-Instruct (Bai et al., 2025b). In addition, we further benchmark our staged-training models against a diverse set of open-weight reasoning VLMs. Specifically, for models built upon Qwen2.5-VL-7B, we include GThinker (Zhan et al., 2025a), MMR1 (Leng et al., 2025), OpenVLThinker (Deng et al., 2025b), R1-OneVision-RL (Yang et al., 2025c), and WeThink (Yang et al., 2025a) as baselines. For models based on Qwen3-VL-8B, we compare against the One-Thinker (Feng et al., 2025b). These baselines represent recent efforts that emphasize visual reasoning, reinforcement learning, or long-chain-of-thought generation, making them strong and relevant comparators for our study. All baseline models are evaluated under their officially released configurations.

**Hyperparameter Setting.** We adopt EasyR1 (Yaowei et al., 2025) as the training framework across all experiments. The system prompt used during training is fixed and provided in Appendix A.4. The maximum response length is set to 2048 tokens, and the sampled group size in Equation 5 is fixed at 5. All experiments are conducted on a server with 8 NVIDIA H200 GPUs.

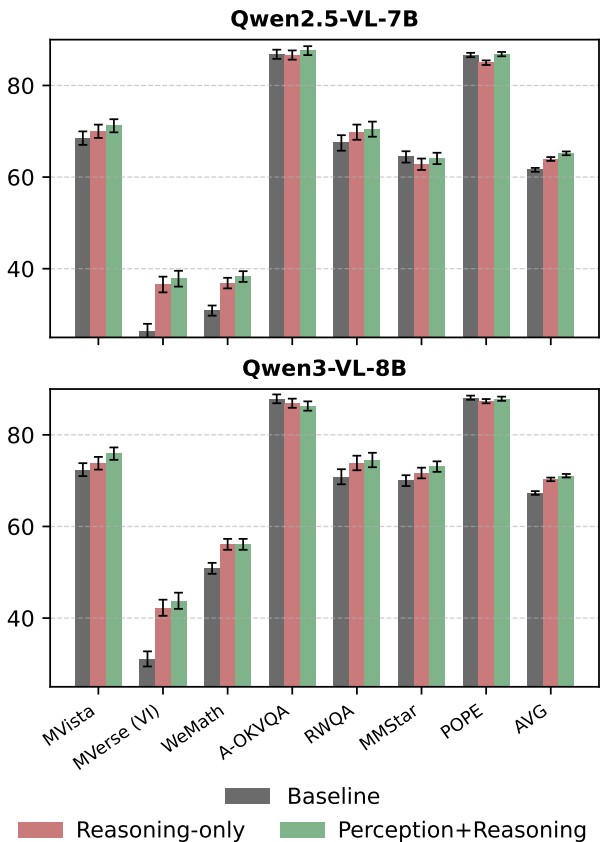

*Figure 3.* **Comparison between the base model, the model trained with reasoning-only, and perception+reasoning data.** Incorporating perception data improves visual math while maintaining perception capabilities. We show standard error bars here, and the exact values are provided in Appendix A.2.

For staged training, visual encoder is enabled for all stages. The number of training steps for the three stages is set to 90, 375, and 465, respectively, ensuring that each stage has the same number of training epochs. For the merged training baseline (Section 3.2.2), the visual encoder is disabled throughout training, following common practice in reasoning-focused post-training (Chen et al., 2025; Yang et al., 2025a). The merged training baseline is trained for 930 steps, matching the total number of training steps used in staged training. More details about the hyperparameter setting are provided in Section A.1.

**Benchmarks.** We evaluate model performance on a comprehensive suite of vision-language benchmarks, covering both visual math reasoning and general visual perception as listed as follow:

- For **visual math reasoning**, we consider MathVista MINI (MVista; Lu et al., 2023), MathVision MINI (MVision; Wang et al., 2024a), MathVerse Vision Intensive subset (MVerse (VI); Zhang et al.,

2024a), and WeMath (Qiao et al., 2025).

- For **perception-oriented**, we include A-OKVQA (Schwenk et al., 2022b), RealWorldQA (RWQA) (xAI, 2024), MMStar (Chen et al., 2024b), and POPE (Li et al., 2023), which assess object recognition, commonsense understanding, real-world perception, and robustness to visual hallucination.

All evaluations are conducted using VLMEvalKit (Duan et al., 2024) as the unified evaluation codebase. We employ Claude-Haiku-4.5 (Anthropic, 2024) as the judge model for all evaluated models and benchmarks.

### 4.2. The Vital Role of Visual Perception in Staged Post-training.

To validate the necessity of visual-dedicated data, we employ a staged, decoupled training pipeline that first establishes a perceptual foundation before introducing complex reasoning. We evaluate this approach through two lenses: an internal ablation on data composition and a broad comparison with strong open-weight baselines.

**The Impact of Visual Perception Data within Staged Training.** We first investigate whether reasoning data alone is sufficient during the post-training stages. We compare three configurations across Qwen2.5-VL-7B and Qwen3-VL-8B: the base models, a reasoning-only staged version (textual and visual), and our proposed incorporation of perception and reasoning data (Figure 3). Across both backbones, the reasoning-only post-training significantly enhances visual math performance; for Qwen2.5-VL-7B, MVerse (VI) and WeMath improve by 10.2% and 6.0%, respectively. However, excluding perception data introduces a "perceptual tax" (Liu et al., 2025). On Qwen2.5-VL-7B, reasoning-only training actually reduces MMStar performance by 1.6%.In contrast, incorporating our visual perception data restores and exceeds base model integrity. By including perception tasks in the staged pipeline, RWQA scores climb to 70.5% (+3.0%) on Qwen2.5-VL-7B and 74.5% (+3.6%) on Qwen3-VL-8B. These results confirm that visual perception data is a fundamental prerequisite for balancing reasoning gains without sacrificing the model's eyes.

**Performance Superiority of Perception-First Training.** To demonstrate the robustness of this decoupled pipeline, we compare our "visual-perception-first" models against specialized open-weight VLMs in Table 1. By prioritizing a solid perceptual foundation before scaling reasoning complexity, we achieve superior results without the trade-offs seen in existing models. In the 7B category, our approach achieves a visual math average of 42.3%, outperforming specialized reasoning baselines like GThinker, OpenVL-

*Table 1.* **Comparison with representative open-weight VLMs (Accuracy %).** Accuracies (%) are reported on individual benchmarks as well as average scores. Best results in each column are highlighted in **bold**, and second-best results are underlined.

| Model | Visual Math | | | | | Perception | | | | | Overall |
|---|---|---|---|---|---|---|---|---|---|---|---|
| | MVista | MVision | MVerse (VI) | WeMath | AVG | A-OKVQA | RWQA | MMStar | POPE | AVG | AVG |
| Qwen2.5-VL-7B | 68.50 | 22.37 | 26.40 | 30.86 | 37.03 | 86.81 | 67.45 | 64.40 | 86.65 | 76.33 | 56.68 |
| GThinker-7B | 69.70 | 23.03 | 36.80 | 35.15 | 41.17 | 86.72 | 69.80 | 64.60 | **88.38** | 77.38 | 59.27 |
| MMR1-7B | 67.40 | 22.04 | 27.03 | 44.67 | 40.29 | 85.07 | 64.31 | 63.67 | 85.78 | 74.71 | 57.50 |
| OpenVLThinker-7B | 70.60 | 22.70 | 36.29 | 35.90 | 41.37 | **88.73** | 69.41 | 63.40 | 80.82 | 75.59 | 58.48 |
| R1-OneVision-RL-7B | 61.70 | 22.04 | 25.25 | 29.71 | 34.68 | 83.58 | 63.40 | 58.13 | 82.11 | 71.81 | 53.24 |
| WeThink-7B | 69.50 | 23.03 | 34.39 | 46.57 | 43.37 | 88.65 | 69.54 | 64.67 | 84.90 | 76.94 | 60.16 |
| Qwen2.5-VL-7B (Staged) | 71.20 | 21.71 | 37.82 | 38.29 | 42.26 | 87.60 | 70.46 | 64.07 | 86.84 | 77.24 | 59.75 |
| Qwen3-VL-8B | 72.40 | 26.32 | 31.09 | 50.86 | 45.17 | 87.86 | 70.85 | 70.00 | 88.11 | 79.21 | 62.19 |
| OneThinker-8B | 75.10 | **33.22** | 41.50 | 54.57 | **51.10** | 86.72 | 71.50 | 70.20 | 86.14 | 78.64 | 64.87 |
| Qwen3-VL-8B (Staged) | **75.90** | 28.62 | **43.78** | **56.10** | **51.10** | 86.29 | **74.51** | **73.07** | 87.88 | **80.44** | **65.77** |

*Table 2.* **Comparison of merged and staged training on the same base VLM across visual math and perception benchmarks (Accuracy %).** Accuracies (%) are reported on individual benchmarks as well as average scores. Best results in each column are highlighted in **bold**, and second-best results are underlined.

| Base Model | Training | Visual Math | | | | | Perception | | | | | Overall |
|---|---|---|---|---|---|---|---|---|---|---|---|---|
| | | MVista | MVision | MVerse (VI) | WeMath | AVG | A-OKVQA | RWQA | MMStar | POPE | AVG | AVG |
| Qwen2.5-VL-7B | Base | 68.50 | **22.37** | 26.40 | 30.86 | 37.03 | 86.81 | 67.45 | **64.40** | 86.65 | 76.33 | 56.68 |
| | Merged | 70.00 | 20.39 | 35.15 | 37.43 | 40.74 | 86.03 | 69.28 | 63.73 | 84.74 | 75.95 | 58.34 |
| | Staged | **71.20** | 21.71 | **37.82** | 38.29 | **42.26** | **87.60** | **70.46** | 64.07 | **86.84** | **77.24** | **59.75** |
| Qwen3-VL-8B | Base | 72.40 | 26.32 | 31.09 | 50.86 | 45.17 | **87.86** | 70.85 | 70.00 | **88.11** | 79.21 | 62.19 |
| | Merged | 73.80 | **28.95** | 40.36 | 55.43 | 49.64 | 85.50 | **75.56** | 70.60 | 87.19 | 79.71 | 64.67 |
| | Staged | **75.90** | 28.62 | **43.78** | **56.10** | **51.10** | 86.29 | 74.51 | **73.07** | 87.88 | **80.44** | **65.77** |

Thinker, and MMR1. Crucially, it maintains a superior average perception score of 77.2%, proving that reasoning capabilities can be scaled more robustly when decoupled from perception.

The advantages are even more pronounced in the Qwen3-VL-8B series. Our staged-training model establishes new state-of-the-art benchmarks for 8B-parameter VLMs, leading in WeMath (56.1%), MathVista (75.9%), MMStar (73.1%), and RealWorldQA (74.5%). These improvements culminate in a record overall average of 65.8%, surpassing both the base model and the reasoning-specialized baseline, OneThinker-8B. These results highlight that explicitly prioritizing visual perception in a staged pipeline is the key to scaling high-performance, general-purpose VLMs.

### 4.3. Beyond One-stage Training: Analyzing Staged Training Paradigms and Ordering

Our training paradigm decomposes VLM post-training into three distinct stages, each targeting a specific capability: visual perception (**Stage 1**), textual reasoning (**Stage 2**), and visual reasoning (**Stage 3**). In this section, we conduct a thorough analysis of this staged training strategy. We begin by comparing it to the conventional single-stage paradigm, where data for all capabilities are combined into one dataset and optimized jointly (**merged training**) as depicted in

Section 3.2.2.

We show that staged training not only delivers higher overall performance but also improves the optimization of visual perception, thereby reducing the cost of reasoning (see Section 4.3.1). In addition, we find that the advantage of staged training depends on the order of the stages: visual perception should be regarded as a more fundamental ability and optimized prior to visual reasoning (see Section 4.3.2).

#### 4.3.1. STAGED VERSUS MERGED TRAINING

**Overall Performance Comparison.** We compare the base models, models with merged training, and those with staged training across visual math perception benchmarks (Table 2). Across both base models, staged training consistently achieves the best overall performance, demonstrating its general effectiveness. For Qwen2.5-VL-7B, staged training improves the average visual math score from 37.0% (base) and 40.7% (merged) to 42.3%, with clear gains on MVerse (26.4% → 37.9%) and WeMath (30.9% → 38.3%). Perception performance is also improved, increasing the average score to 77.2%, compared to 76.3% (base) and 76.0% (merged), resulting in the highest overall score of 59.8%.

Similar trends are observed for models with the Qwen3-VL-8B backbone. Staged training outperforms both base and merged training on visual math, improving the average score

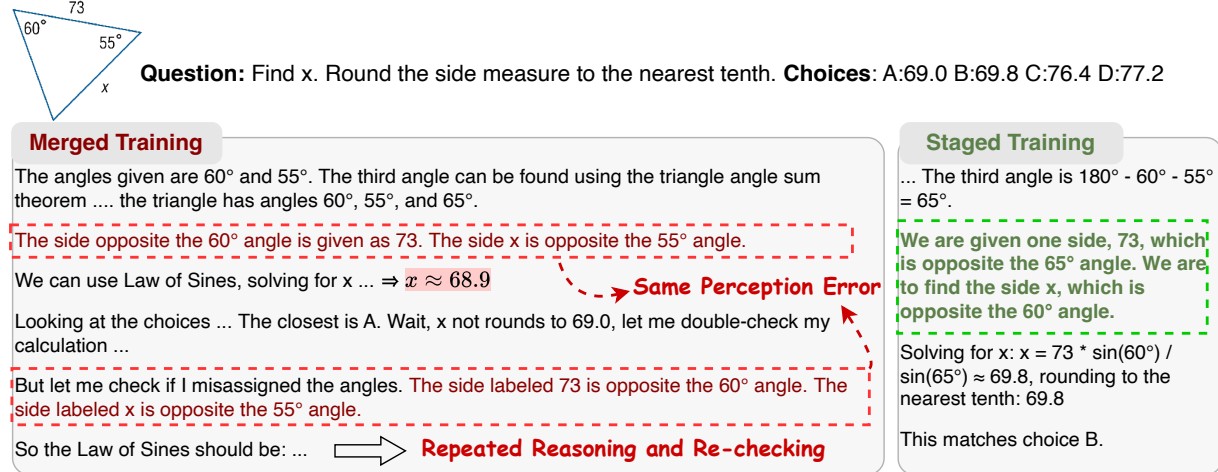

**Question:** Find x. Round the side measure to the nearest tenth. **Choices**: A:69.0 B:69.8 C:76.4 D:77.2

**Merged Training**

The angles given are 60° and 55°. The third angle can be found using the triangle angle sum theorem .... the triangle has angles 60°, 55°, and 65°.

The side opposite the 60° angle is given as 73. The side x is opposite the 55° angle.

We can use Law of Sines, solving for x ... ⇒ $x \approx 68.9$

**Same Perception Error**

Looking at the choices ... The closest is A. Wait, x not rounds to 69.0, let me double-check my calculation ...

But let me check if I misassigned the angles. The side labeled 73 is opposite the 60° angle. The side labeled x is opposite the 55° angle.

So the Law of Sines should be: ... ⟹ **Repeated Reasoning and Re-checking**

**Staged Training**

... The third angle is 180° - 60° - 55° = 65°.

We are given one side, 73, which is opposite the 65° angle. We are to find the side x, which is opposite the 60° angle.

Solving for x: x = 73 * sin(60°) / sin(65°) ≈ 69.8, rounding to the nearest tenth: 69.8

This matches choice B.

*Figure 4.* **Case Study between Staged and Merged Training Models.** The staged training model generates concise reasoning with correct perception.

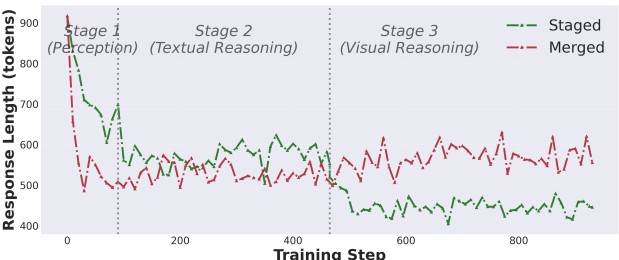

*Figure 5.* **Staged Training Reduces the Response Length for Visual Reasoning.** For the Qwen3-VL-8B model, we plot the average response length on the validation set over training steps, comparing the staged and merged training strategies.

from 45.2% to 51.1% and achieving the best perception (average 80.4%). Consequently, staged training attains the highest overall score (65.8%) among all variants. To further verify the generality of staged training beyond the Qwen family, we evaluate on InternVL3.5-8B and InternVL3-8B (Appendix A.5). Staged training consistently outperforms merged training across both InternVL architectures, with overall gains of +0.95% and +3.77%, respectively, confirming that the benefit of decoupling perception and reasoning generalizes across different VLM backbones. We further validate statistical robustness by averaging over three independent runs across 15 benchmarks (Appendix A.6); staged training wins on 14/15 benchmarks for Qwen3-VL-8B.

In addition, we employ Claude-4.5-Haiku to assess perception errors in the model's reasoning, with the complete prompt provided in the Appendix A.4. We randomly selected 40 judgments from Claude-4.5-Haiku and manually verified whether it correctly identifies visual perception errors. The validation shows that 33/40 (82.5%) of the samples match human judgments, suggesting that the Claude model is reliable for detecting visual perception errors. For the

Qwen3-VL-8B model, 857 out of 3044 samples from the MVista, MVision, and WeMath benchmarks are identified as having perception errors. After merged training, this count drops to 805, and staged training further decreases it to 781, indicating that explicitly decoupling perception and reasoning during training leads to more effective and robust VLM performance.

**Staged Training Leads to Better Perception and Shorter Thinking Costs.** Figure 5 shows the average response length during training for Qwen3-VL-8B under staged and merged training. While both approaches start with long responses, the staged model gradually reduces its response length as perception training progresses. During Stage 2, staged training maintains response lengths comparable to merged training, indicating that shorter outputs are not caused by suppressed reasoning. A clear divergence appears in Stage 3, where the staged model produces responses that are 20.8% shorter than those from merged training (average length 445 tokens *v.s.* 562 tokens across the validation set), while achieving higher math reasoning accuracy, as shown in Table 2 (51.1% *v.s.* 49.6%). This reduction is consistent at test time: on four visual math benchmarks, staged training produces 6.6–12.6% shorter responses (Appendix A.7).

Figure 4 provides a detailed comparison between merged and staged training. Under merged training, the model incorrectly assigns the side of length 73 to the wrong angle. This perceptual error persists across repeated image checks, leading to long and repeated reasoning traces without resolving the inconsistency. In contrast, the staged-trained model correctly identifies the geometric relationships at the outset. With accurate perception, the subsequent reasoning becomes concise and directly yields the correct answer, explaining the shorter response lengths observed in Figure 5.

*Table 3.* **Effect of stage order on visual math and perception performance (Accuracy %).** Best results in each column are highlighted in **bold**. We compare different stage orders for staged training on Qwen2.5-VL-7B and Qwen3-VL-8B. Stage 1→2→3 (perception → textual reasoning → visual reasoning) and Stage 2→1→3 achieve comparable and consistently strong performance, while reversing the order to Stage 3→2→1 leads to clear degradation in both visual math and perception metrics.

| Model / Stage Order | Visual Math | | | | | Perception | | | | | Overall |
|---|---|---|---|---|---|---|---|---|---|---|---|
| | MVista | MVision | MVerse (VI) | WeMath | AVG | A-OKVQA | RWQA | MMStar | POPE | AVG | AVG |
| Qwen2.5-VL-7B (Base) | 68.50 | 22.37 | 26.40 | 30.86 | 37.03 | 86.81 | 67.45 | 64.40 | 86.65 | 76.33 | 56.68 |
| Qwen2.5-VL-7B ( 1→2→3) | **71.20** | 21.71 | **37.82** | **38.29** | 42.26 | **87.60** | **70.46** | 64.07 | **86.84** | **77.24** | **59.75** |
| Qwen2.5-VL-7B ( 2→1→3) | 71.70 | 23.36 | 38.20 | 38.38 | **42.91** | 86.46 | 69.54 | **64.67** | 84.54 | 76.30 | 59.61 |
| Qwen2.5-VL-7B ( 3→2→1) | 66.60 | **24.34** | 32.99 | 26.86 | 37.70 | 79.48 | 68.63 | 62.87 | 85.70 | 74.17 | 55.93 |
| Qwen3-VL-8B (Base) | 72.40 | 26.32 | 31.09 | 50.86 | 45.17 | 87.86 | 70.85 | 70.00 | **88.11** | 79.21 | 62.19 |
| Qwen3-VL-8B ( 1→2→3) | **75.90** | **28.62** | **43.78** | 56.10 | **51.10** | 86.29 | 74.51 | 73.07 | 87.88 | 80.44 | 65.77 |
| Qwen3-VL-8B ( 2→1→3) | 74.90 | **28.62** | 43.65 | 55.81 | 50.75 | 86.72 | **76.21** | **73.33** | 87.17 | **80.86** | **65.80** |
| Qwen3-VL-8B ( 3→2→1) | 75.20 | 26.64 | 40.86 | **57.43** | 50.03 | 84.45 | 74.77 | 71.33 | 87.65 | 79.55 | 64.79 |

*Table 4.* **Effect of reinforcement learning versus supervised fine-tuning for Stage 1 perception training (Accuracy %).** Best results in each column are highlighted in **bold**. Across both Qwen2.5-VL-7B and Qwen3-VL-8B, RLVR consistently yields higher perception accuracy and leads to stronger downstream visual math performance, resulting in improved overall accuracy.

| Model / Stage 1 Method | Visual Math | | | | | Perception | | | | | Overall |
|---|---|---|---|---|---|---|---|---|---|---|---|
| | MVista | MVision | MVerse (VI) | WeMath | AVG | A-OKVQA | RWQA | MMStar | POPE | AVG | AVG |
| Qwen2.5-VL-7B (RLVR) | **71.20** | **21.71** | **37.82** | **38.29** | **42.26** | **87.60** | **70.46** | **64.07** | **86.84** | **77.24** | **59.75** |
| Qwen2.5-VL-7B (SFT) | 66.40 | 18.75 | 32.87 | 30.10 | 37.03 | 84.63 | 70.20 | 62.60 | 85.26 | 75.67 | 56.35 |
| Qwen3-VL-8B (RLVR) | **75.90** | 28.62 | **43.78** | **56.10** | **51.10** | **86.29** | **74.51** | **73.07** | **87.88** | **80.44** | **65.77** |
| Qwen3-VL-8B (SFT) | 74.70 | **32.24** | 42.89 | 54.48 | 51.08 | 85.94 | 72.29 | 72.60 | 86.00 | 79.21 | 65.14 |

### 4.3.2. STAGE ORDER MATTERS

Table 3 analyzes the impact of different stage orders on visual math and perception performance. Across both Qwen2.5-VL-7B and Qwen3-VL-8B, we observe that the order of staged training plays a critical role in determining final model performance. For both model series, Stage 1→2→3 (visual perception → textual reasoning → visual reasoning) consistently yields strong and balanced performance across visual math and perception benchmarks. Exchanging the first two stages (Stage 2→1→3) results in comparable average scores for math and perception. For the Qwen2.5-VL-7B model, these two training orders achieve 42.3% *v.s.* 42.9% average scores across visual math benchmarks and 77.2% *v.s.* 76.3% on visual perception, suggesting that visual perception and textual reasoning function as complementary foundational capabilities that can be learned in either order before visual reasoning.

In contrast, reversing the order to Stage 3→2→1 leads to a clear degradation in performance. For Qwen2.5-VL-7B, the visual math average score drops from over 42% to 37.7%, and the visual perception average decreases to 74.2%, approaching the base model level. A similar trend is observed for Qwen3-VL-8B, where Stage 3→2→1 underperforms both Stage 1→2→3 and Stage 2→1→3 in overall accuracy (64.8% *v.s.* 65.8% *v.s.* 65.8%). This indicates that prematurely training visual reasoning entangles perception and

reasoning before either capability is sufficiently established.

Taken together, these findings indicate that staged training is not just about isolating different capabilities but also about acquiring them in a suitable sequence. Visual perception, as a fundamental skill, should be solidified before visual reasoning to maximize the effectiveness of staged training.

### 4.4. RLVR is More Effective than SFT for Perception Training

Caption-based supervised fine-tuning (SFT) is a widely adopted approach for aligning LLMs to the vision modality (Liu et al., 2024a; Chen et al., 2024a; 2023; Ogezi & Shi, 2025; Sun et al., 2024), as it provides direct supervision on image-text correspondence. To examine whether this approach is suitable for enhancing perception at the post-training stage, we compare caption-based SFT with our RLVR approach in Stage 1 (visual perception) training, followed by the same training setups in subsequent stages.

As shown in Table 4, RLVR consistently outperforms SFT across both Qwen2.5-VL-7B and Qwen3-VL-8B. In particular, RLVR leads to higher average visual perception scores (e.g., 77.2% *v.s.* 75.7% on Qwen2.5-VL-7B and 80.4% *v.s.* 79.2% on Qwen3-VL-8B), and these improvements translate into stronger visual math performance. Notably, employing RLVR in visual perception training leads to an 8.2% performance gain on Qwen2.5-VL-7B and the WeMath bench-

*Table 5.* **Effect of combining capability-based and difficulty-based curricula on Qwen3-VL-8B (Accuracy %).** Best results in each column are highlighted in **bold**.

| Curriculum | MVision | MVerse (VO) | WeMath | DynaMath | RWQA | CV-Bench | V*Bench | AVG |
|---|---|---|---|---|---|---|---|---|
| None (Merged) | 28.95 | 36.68 | 55.43 | 54.15 | 75.56 | 78.53 | 80.63 | 58.56 |
| Capability | 28.62 | 39.97 | 56.10 | 61.14 | 74.51 | 79.62 | 83.77 | 60.53 |
| Difficulty | 24.67 | 40.86 | 54.10 | 67.07 | 72.68 | 79.36 | 83.77 | 60.36 |
| Capability+Difficulty | **33.22** | **41.75** | **57.43** | **67.49** | **75.82** | **80.95** | **84.29** | **62.99** |

mark, increasing the average visual math score from 37.0% to 42.3%. While SFT occasionally achieves competitive results on individual benchmarks (e.g., MathVision), RLVR provides more stable and consistent gains across both perception and reasoning metrics. These results suggest that although caption-based SFT has been proven effective for vision-language alignment, RLVR offers a stronger training signal for perception by explicitly penalizing unsupported or hallucinated visual interpretations. As a result, RL-based perception training leads to more reliable visual grounding and improved downstream reasoning performance.

### 4.5. Complementarity with Difficulty-Based Curriculum

Our staged training can be viewed as a *capability-based curriculum*—organizing training by functional role (perception → reasoning) rather than by sample difficulty. To investigate whether this new curriculum dimension is complementary to traditional difficulty-based ordering, we compare four training configurations on Qwen3-VL-8B: merged training (no curriculum), capability-only (our staged training), difficulty-only (samples ordered by hardness within merged training), and the combination of both (difficulty ordering applied within each capability stage). To estimate sample difficulty, we sample 16 answers per question from Qwen3-VL-8B with temperature 1.0 and compute the average pass rate as a difficulty score. Training samples are then ranked from easy (high pass rate) to hard (low pass rate). For *difficulty-only*, we apply this ranking to the entire merged dataset; for *capability+difficulty*, we apply the ranking *within* each of the three capability stages and train the stages in our standard order (perception → textual reasoning → visual reasoning), with easy samples preceding hard ones in every stage. We evaluate on a diverse set including MathVerse Vision Only subset (MVerse (VO); Zhang et al., 2024a), DynaMath (Zou et al., 2025), CV-Bench (Zhu et al., 2025), and V*Bench (Wu & Xie, 2024).

As shown in Table 5, both capability-based and difficulty-based curricula individually improve over merged training (60.53% and 60.36% *v.s.* 58.56%). Crucially, combining the two dimensions yields a further gain to 62.99%, surpassing either curriculum alone by over 2%. This demonstrates that capability-based staging and difficulty-based ordering address orthogonal aspects of training optimization and can be effectively composed for additive improvements.

## 5. Discussion and Conclusion

In this work, we establish that visual perception is a dominant limiting factor for visual reasoning in VLMs and that longer reasoning alone cannot compensate for perceptual errors. Motivated by this insight, we introduce a staged post-training paradigm that decouples VLM capabilities into visual perception, textual reasoning, and visual reasoning stages. This decoupled approach consistently outperforms unified training pipelines across four model architectures while producing shorter reasoning traces, and we demonstrate that RLVR provides a more effective training signal than caption-based SFT for perception optimization.

Conceptually, our staged framework can be viewed as *capability-dimension curriculum learning*—a framework that complements existing difficulty-dimension curricula (Zhang et al., 2025; Liu et al., 2024b). Rather than scaling tasks by difficulty, we structure training by functional roles, and show that combining both curriculum dimensions yields further additive improvements (Section 4.5). This suggests a promising direction for multidimensional training trajectories in future VLM post-training.

**Limitations.** Our study has several limitations. First, all experiments are conducted at the 7–8B parameter scale; validation on larger models (32B+) remains future work. Second, our perception data pipeline relies on the availability of fine-grained image captions, which may limit applicability to domains without such resources. Third, our three-stage decomposition may not represent the finest granularity of capability separation; exploring more fine-grained stage decompositions is an interesting direction.

## Impact Statement

This paper presents work whose goal is to advance the field of Machine Learning. There are many potential societal consequences of our work, none of which we feel must be specifically highlighted here.

## Acknowledgements

This work was partially funded by an unrestricted gift from Google.

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

# A. Appendix

## A.1. Detailed Hyperparameter Setting

We provide the full hyperparameter in Table 6 For all remaining training parameters not listed in the table, we follow the default settings of EasyR1 (Yaowei et al., 2025) to ensure a controlled comparison and reproducibility.

*Table 6.* **Key hyperparameters used in our Stage-3 training.**

| Hyperparameter | Value |
|---|---|
| Max prompt length | 2048 |
| Actor dtype | bf16 |
| Actor optimizer | adamw_bf16 |
| Actor micro-bsz (update) | 16 |
| Actor micro-bsz (experience) | 64 |
| Offload params / optim | False / False |
| Rollout GPU mem util. | 0.7 |
| Tensor parallel size | 1 |
| Reward type | sequential |
| GPUs per node | 8 |

## A.2. More Experimental Results

**Ablation of each training stage.**

*Table 7.* **Ablation study of different staged training combinations on Qwen3-VL-8B (Accuracy %).**

| Training Stages | MVista | MVision | WeMath | A-OKVQA | RWQA | POPE | AVG |
|---|---|---|---|---|---|---|---|
| Base Model | 72.40 | 26.32 | 50.86 | 87.86 | 70.85 | 88.11 | 66.07 |
| Stage 3 | 73.90 | 26.64 | 56.10 | 86.03 | 73.59 | 87.69 | 67.33 |
| Stage 1→3 | 73.90 | 29.28 | 58.76 | 86.55 | 73.59 | 87.55 | 68.27 |
| Stage 2→3 | 73.80 | 27.30 | 56.10 | 86.90 | 73.86 | 87.35 | 67.55 |
| Stage 1→2→3 | 75.90 | 28.62 | 56.10 | 86.29 | 74.51 | 87.88 | 68.22 |

The ablation results in Table 7 further verify the critical role of visual perception training. Compared with applying Stage 3 (visual reasoning) alone, introducing visual perception-oriented Stage 1 before Stage 3 yields clear gains on visual math benchmarks, with MVision improving from 26.64% to 29.28% and WeMath from 56.10% to 58.76%, and the overall average increasing from 67.33% to 68.27%. In contrast, directly adding Stage 2 before Stage 3 leads to only marginal changes (AVG: 67.33% *v.s.* 67.68%), indicating that reasoning-oriented improvements largely saturate when perception remains weak. Moreover, incorporating Stage 1 prior to both Stage 2 and Stage 3 further enhances performance over Stage 2→3, particularly on MVista (75.90% *v.s.* 73.80%). Together, these findings demonstrate that visual perception constitutes a dominant bottleneck in current VLMs, and explicitly strengthening perception is a prerequisite for unlocking effective downstream reasoning improvements.

**Impact of Training Vision Encoder.**

*Table 8.* **Effect of vision encoder freezing strategies under staged and merged training (Accuracy %).** *Mixed* denotes the strategy used in the main paper, where the vision encoder is frozen in Stage 2 but unfrozen in Stage 1 and Stage 3.

| Model / Training | Vision Encoder | MVista | MVision | WeMath | A-OKVQA | RWQA | POPE | AVG |
|---|---|---|---|---|---|---|---|---|
| Qwen2.5-VL-7B / Staged | Mixed | 71.20 | 21.71 | 38.29 | 87.60 | 70.46 | 86.84 | 62.68 |
| Qwen2.5-VL-7B / Staged | All Freeze | 70.00 | 20.72 | 38.38 | 86.90 | 69.02 | 86.33 | 61.89 |
| Qwen2.5-VL-7B / Staged | All Open | 70.50 | 22.37 | 38.67 | 87.16 | 69.93 | 86.08 | 62.45 |
| Qwen2.5-VL-7B / Merged | All Freeze | 70.90 | 20.39 | 36.67 | 85.59 | 70.07 | 84.35 | 61.33 |
| Qwen2.5-VL-7B / Merged | All Open | 70.00 | 20.39 | 37.43 | 86.03 | 69.28 | 84.74 | 61.31 |
| Qwen3-VL-8B / Staged | Mixed | 75.90 | 28.62 | 56.10 | 86.29 | 74.51 | 87.88 | 68.22 |
| Qwen3-VL-8B / Staged | All Freeze | 75.20 | 30.26 | 56.57 | 86.11 | 76.21 | 87.42 | 68.63 |
| Qwen3-VL-8B / Staged | All Open | 75.30 | 31.91 | 54.67 | 86.55 | 74.90 | 87.18 | 68.42 |
| Qwen3-VL-8B / Merged | All Freeze | 74.30 | 25.99 | 54.38 | 85.33 | 73.86 | 87.40 | 66.88 |
| Qwen3-VL-8B / Merged | All Open | 73.80 | 28.95 | 55.43 | 85.50 | 75.56 | 87.19 | 67.74 |

*Table 9.* **Comparison between the base model, the model trained with reasoning-only, and perception+reasoning data (Accuracy %).** Incorporating perception data improves visual math while maintaining perception capabilities. We show standard error bars in Figure 2, and the exact values are provided in this table.

| Model | MVista | MVerse (VI) | WeMath | A-OKVQA | RWQA | MMStar | POPE | AVG |
|---|---|---|---|---|---|---|---|---|
| Qwen2.5-VL-7B | 68.50 | 26.40 | 30.86 | 86.81 | 67.45 | 64.40 | 86.65 | 61.58 |
| Qwen2.5-VL-7B (Perception+Reasoning) | 71.20 | 37.82 | 38.29 | 87.60 | 70.46 | 64.07 | 86.84 | 65.18 |
| Qwen2.5-VL-7B (Reasoning-only) | 70.00 | 36.55 | 36.86 | 86.64 | 69.80 | 62.80 | 84.97 | 63.95 |
| Qwen3-VL-8B | 72.40 | 31.09 | 50.86 | 87.86 | 70.85 | 70.00 | 88.11 | 67.31 |
| Qwen3-VL-8B (Perception+Reasoning) | 75.90 | 43.78 | 56.10 | 86.29 | 74.51 | 73.07 | 87.88 | 71.08 |
| Qwen3-VL-8B (Reasoning-only) | 73.80 | 42.26 | 56.10 | 86.90 | 73.86 | 71.67 | 87.35 | 70.28 |

Table 8 compares different vision encoder freezing strategies under both staged and merged training. Across settings, varying the vision encoder between fully frozen, fully trainable, and the proposed mixed strategy leads to relatively small performance differences, suggesting that encoder freezing alone is not a dominant factor governing final performance. In contrast, staged training consistently outperforms merged training under comparable encoder configurations on both Qwen2.5-VL-7B and Qwen3-VL-8B. For example, on Qwen2.5-VL-7B, staged training models achieve higher average accuracy (up to 62.68%) than their merged counterparts (around 61.3%), while on Qwen3-VL-8B, staged training reaches 68.22%–68.63% compared to 66.88%–67.74% under merged training. These consistent gains across architectures indicate that the staged training paradigm itself, rather than specific encoder freezing heuristics, is the primary driver of performance improvements.

**Exact Values in Section 4.2.**

Table 9 reports the exact values corresponding to Figure 2. Across both Qwen2.5-VL-7B and Qwen3-VL-8B, incorporating perception data consistently yields larger gains on visual math benchmarks compared to reasoning-only training. For instance, on Qwen3-VL-8B, perception+reasoning improves MVerse (VI) from 42.26% to 43.78% and MVista from 73.80% to 75.90%, while achieving comparable performance on A-OKVQA and POPE. A similar trend is observed in Qwen2.5-VL-7B, where perception+reasoning outperforms reasoning-only on WeMath (38.29% *v.s.* 36.86%) and MVerse (VI) (37.82% *v.s.* 36.55%). These results indicate that strengthening visual perception directly translates into improved visual reasoning without degrading general perception capabilities.

### A.3. Visual Perception Data Example

Here, we include two representative generated visual perception examples (Figure 6). The first requires robust *object detection and counting under low-light conditions* by identifying seven streetlamps and their reflections on the river surface. The second targets *fine-grained visual attribute discrimination*, where the model must infer the most recently painted letter in a weathered graffiti word based on color intensity and paint texture.

Together, these examples illustrate that our generated perception data explicitly exercises core visual competencies such as object counting, reflection understanding, fine-grained appearance comparison, and material aging cues—capabilities that are often bottlenecks in downstream visual reasoning tasks.

### A.4. Prompt Settings

In this section, we provide all the prompts used in our experiments, including the prompt for (a) generating visual perception question-answering data (Figure 7); (b) assessing visual perception errors in the model's reasoning (Figure 8); and (c) the system prompt used for model training (Figure 9).

### A.5. Extended Benchmark Results Across Four Model Families

Table 10 presents a comprehensive evaluation across four model families on ten extended benchmarks, including Dyna-Math (Zou et al., 2025), HallusionBench (Guan et al., 2024), BLINK (Fu et al., 2024), VisOnlyQA (Kamoi et al., 2024), V*Bench (Wu & Xie, 2024), and CV-Bench (Zhu et al., 2025). Staged training consistently outperforms merged training across all architectures: InternVL3-8B shows the largest gain (+3.77% overall), followed by Qwen3-VL-8B (+3.37%), Qwen2.5-VL-7B (+1.62%), and InternVL3.5-8B (+0.95%). Notably, for InternVL3-8B, staged training improves WeMath from 25.05% to 34.95% (+9.90%), demonstrating that the benefit of decoupling perception and reasoning is especially

*Table 10.* **Comprehensive comparison of base, merged, and staged training across four model families on extended benchmarks (Accuracy %).** Best results within each model family are highlighted in **bold**.

| Base Model | Training | MVista | MVerse (VO) | MVerse (VI) | WeMath | DynaMath | MMStar | Hallusion | BLINK | VisOnlyQA | VStarBench | AVG |
|---|---|---|---|---|---|---|---|---|---|---|---|---|
| | Base | 60.70 | 11.93 | 5.08 | 40.10 | 44.85 | 43.27 | 39.88 | 46.66 | 36.89 | 43.98 | 37.33 |
| InternVL3.5-8B | Merged | 69.40 | 32.11 | 32.61 | 48.38 | **62.34** | **64.40** | 48.55 | 56.13 | 50.89 | **62.83** | 52.76 |
| | Staged | **70.30** | **33.76** | **34.26** | **49.90** | 62.83 | 61.60 | **53.47** | **57.71** | **52.00** | 61.26 | **53.71** |
| | Base | 18.10 | 17.13 | 22.34 | 3.14 | 13.15 | 46.33 | 30.06 | **48.55** | 41.56 | 56.54 | 29.69 |
| InternVL3-8B | Merged | 60.70 | 25.25 | 30.46 | 25.05 | 45.79 | 51.87 | 30.35 | 48.45 | 40.22 | **61.26** | 41.94 |
| | Staged | **65.40** | **29.70** | **32.99** | **34.95** | **52.42** | **53.00** | **36.42** | 46.87 | **46.22** | 59.16 | **45.71** |
| | Base | 68.40 | 24.11 | 25.00 | 30.86 | 51.54 | 63.67 | **40.46** | 55.39 | 49.56 | **76.96** | 48.60 |
| Qwen2.5-VL-7B | Merged | 69.75 | 29.57 | 34.23 | **37.24** | 52.89 | 63.33 | 36.56 | 54.60 | 47.95 | 76.83 | 50.30 |
| | Staged | **71.45** | **32.93** | **38.13** | 36.88 | **54.03** | **64.79** | 39.95 | **55.71** | **48.67** | 76.70 | **51.92** |
| | Base | 72.40 | 26.90 | 31.09 | 50.86 | **66.43** | 70.00 | 32.95 | **68.39** | 61.78 | **83.77** | 56.46 |
| Qwen3-VL-8B | Merged | 73.80 | 36.68 | 40.36 | 55.43 | 54.15 | 70.60 | 53.18 | 61.34 | 61.56 | 80.63 | 58.77 |
| | Staged | **75.90** | **39.97** | **43.78** | **56.10** | 61.14 | **73.07** | **59.54** | 64.12 | **64.00** | **83.77** | **62.14** |

impactful for weaker base models. These results confirm that our staged training paradigm generalizes beyond the Qwen family to architecturally distinct VLMs.

## A.6. Statistical Robustness: Three-Run Averaged Results

*Table 11.* **Three-run averaged results for Qwen3-VL-8B and Qwen2.5-VL-7B (Accuracy %).** Staged training consistently outperforms merged training across all averaged benchmarks, demonstrating statistical robustness. Best results in each row pair are in **bold**.

| Model | MVista | MVision | MVerse (VO) | MVerse (VI) | WeMath | DynaMath | A-OKVQA | RWQA | MMStar | Hallusion | POPE | BLINK | CV-Bench | VisOnlyQA | VStarBench | AVG |
|---|---|---|---|---|---|---|---|---|---|---|---|---|---|---|---|---|
| Qwen2.5-VL-7B (Staged) | **71.50** | **20.94** | **32.95** | **38.28** | 37.02 | **54.05** | **87.07** | 69.54 | **64.82** | **39.98** | **87.06** | **55.74** | **75.95** | **48.82** | **76.79** | **57.37** |
| Qwen2.5-VL-7B (Merged) | 69.63 | 19.52 | 29.40 | 34.01 | **37.21** | 52.84 | 85.85 | **70.15** | 63.31 | 36.42 | 84.67 | 54.57 | 74.72 | 47.85 | 76.62 | 55.78 |
| Qwen3-VL-8B (Staged) | **76.20** | **28.84** | **40.01** | **43.23** | **56.86** | **68.67** | **86.99** | **74.08** | **73.15** | **55.88** | **87.63** | **65.40** | **80.70** | **64.08** | **85.51** | **65.82** |
| Qwen3-VL-8B (Merged) | 72.93 | 26.86 | 35.91 | 38.88 | 52.98 | 66.58 | 85.42 | 73.77 | 70.20 | 50.97 | 87.18 | 62.97 | 78.31 | 62.15 | 80.28 | 63.03 |

Table 11 reports results averaged over three independent evaluation runs. Staged training outperforms merged training on 14/15 benchmarks for Qwen3-VL-8B (+2.79% overall AVG) and 12/15 benchmarks for Qwen2.5-VL-7B (+1.59% overall AVG). The few benchmarks where merged training leads (e.g., WeMath and RWQA for Qwen2.5-VL-7B) show differences within 0.6%, well within noise. These results confirm that the improvements from staged training are statistically robust and not artifacts of evaluation variance.

## A.7. Response Length on Test Sets

*Table 12.* **Average response length (tokens) on visual math test sets for Qwen3-VL-8B.** Staged training produces shorter responses across all benchmarks while achieving higher accuracy (Table 2).

| Model | MathVista | MathVision | MathVerse (VO) | WeMath |
|---|---|---|---|---|
| Staged | 1325.89 | 2930.41 | 1541.89 | 1745.69 |
| Merged | 1420.30 | 3163.41 | 1764.93 | 1906.07 |
| Reduction | −6.6% | −7.4% | −12.6% | −8.4% |

Table 12 shows that staged training produces 6.6–12.6% shorter responses across all visual math test benchmarks compared to merged training, consistent with the training-time observation in Figure 5. Combined with the higher accuracy achieved by staged training, this confirms that stronger perception reduces the need for excessive reasoning and repeated image re-checking.

## Visual perception data examples

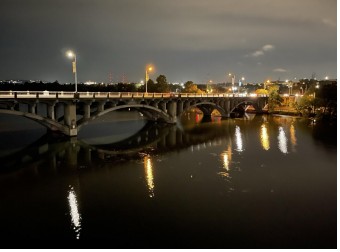

Question: How many utility streetlamps are **reflected** in the river?
Options: A: "Five", B: "Six", C: "Seven", D: "Eight"
Caption: An outdoor, nigh time view of an arched vehicular bridge over a river. The city lights beyond the bridge are making it less than dark. The river makes up the bottom half of the frame. It is angled from the bottom right corner slightly up toward the left. The bridge has four large gradual arches across the river from the left middle of the frame to the right.\"T\" shaped pillars are on top of the arches holding up the bridge. A well lit guardrail runs across the length of the bridge on both sides. Seven utility streetlamps are on top of the bridge. They highlight the guardrail and bridge. All the streetlamps are reflected in the river. The reflections run vertically and are elongated. They have rough edges of shimmer. The reflections are white and yellow. The top third of the image is a cloudy night sky.

---

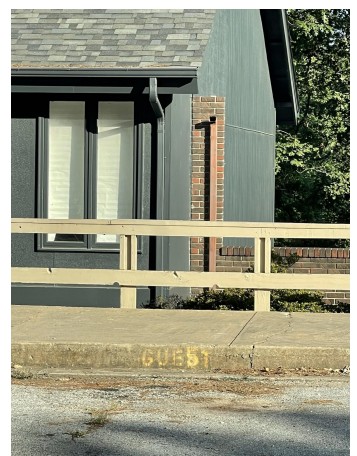

Question: Which letter in the word 'GUEST' appears to have **been painted more recently** than the others?
Options: A: "G", B: "U", C: "S", D: "T"
Caption: The word \"GUEST\" is spray painted on the side of a sidewalk that sits outside a building that is painted pine green. The wording has been weathered with time. The \"G\",\"U\",\"E\" and \"T\" are fainter than the letter \"S\" that seems to have been painted over in recent years. The paint on the \"S\" has dried running drips. The building is seen from the side view. There are two windows to the left of the drainage gutter coming down from the roof. The gutter pipe is painted in the same pine green color as the walls of the house. The two windows to the left of the gutter are tall and thin and have single panes of glass that match the dimensions of the windows. One of the windows to the left has thin spaces around the circumference hinting that the window can be opened. To the right of the gutter is a thin red brick structure that has a detached piece of red gutter with no connecting pieces to the roof of the house. The red brick structure connects to a lower brick wall that runs to the right of the image. The pavement in the parking space just before the side is marbled and gray. There is a layer of dirt and pollen that has spread around the pavement and dirtied it. The side of the sidewalk is dirtier on the left side of the word \"GUEST.\" A cream railing made from wide wooden planks blocks off the access to the green building.

*Figure 6.* **Example of synthesized visual perception data.**

## Prompt for generating visual perception training data

messages = [ {"role": "user", "content": """ You are an assessment-item writer.
Your job is to create a single multiple-choice question (MCQ) that can only be answered by carefully understanding the image described below.

\# Image description
{description}

\# Your task
1. Read the description attentively.
2. Write one question that targets either
    - a spatial relationship in the image (e.g., relative positions, directions, sizes, distances), or
    - a fine detail visible in the scene (e.g., colors, numbers, small objects, text).
3. Create four answer options (keys A, B, C, D) that all sound plausible but only one is correct.
    - Make the distractors non-trivial: a casual glance at the description should not reveal the answer.
    - Match the wording and specificity of the correct answer.
4. Mark the correct option with the field answer, whose value is the single capital letter A / B / C / D.
5. Return your result only as JSON inside a Markdown code block fenced with ```json—no additional text.

\# Output schema
```json
{{
  "question": "string",
  "options": {{
    "A": "string",
    "B": "string",
    "C": "string",
    "D": "string"
  }},
  "answer": "A|B|C|D"
}}"""}
]

*Figure 7.* **Prompt for generating visual perception question-answer pairs.**

### Prompt for assessing visual perception error

messages = [ {"role": "user", "content": """ You are an expert at analyzing VLM (Vision-Language Model) outputs for perception errors.

A perception error occurs when the model incorrectly interprets visual information from an image, including:
1. **Object Identification Errors**: Misidentifying objects (e.g., calling a cat a dog)
2. **Attribute Errors**: Wrong attributes like color, size, shape, texture (e.g., saying a red apple is green)
3. **Spatial Relation Errors**: Incorrect spatial relationships (e.g., saying object A is above B when it's below)
4. **Counting Errors**: Wrong number of objects
5. **Text Reading Errors**: Misreading text/numbers shown in the image

Given the following question and the model's prediction, determine if there is a perception error.

**Question:**
{question}

**Model's Prediction:**
{prediction}

**Ground Truth Answer:**
{answer}

Analyze the model's prediction and determine:
1. Does the prediction contain any perception error? (YES/NO)
2. If YES, what type of perception error(s)? Briefly describe.

Respond in the following JSON format only:
{{
    "has_perception_error": true/false,
    "error_types": ["list of error types if any"],
    "explanation": "brief explanation"
}}"""}
]

*Figure 8.* **Prompt for assessing visual perception errors in VLM's reasoning.**

### Prompt for model training

messages = [ {"role": "system", "content": """You FIRST think about the reasoning process as an internal monologue and then provide the final answer. The reasoning process MUST BE enclosed within <think> </think> tags. The final answer MUST BE put in \boxed{}. i.e. <thinking> reasoning here </thinking> \boxed{final answer here} """}
]

*Figure 9.* **System prompt used in our experiments.**

