# OpenReview forum: "From Seeing to Thinking: Decoupling Perception and Reasoning Improves Post-Training of Vision-Language Models"
_ICML.cc/2026/Conference — ICML 2026 regular_

### Official Review · Reviewer_jL8w · 2026-03-11

**Soundness:** 3
**Presentation:** 3
**Significance:** 2
**Originality:** 2
**Overall Recommendation:** 5
**Confidence:** 3

**Summary:**

This paper tackles the challenge of multimodal perception in visual reasoning tasks.

First, the authors analyse VLMs errors on 3 visual math datasets, highlighting that 87% are due to perception errors (with Qwen3-VL-8B).
Hence, they argue that visual perception requires further targeted optimization with specialized data.

They propose a post-training framework in three stages: visual perception, textual reasoning, and visual reasoning.
For stage 1 (perception), they create a dedicated dataset for RLVR training, prompting an LLM  (Qwen2.5-72B) to generate a set of perception-focused QA pairs from a set of captions (the captions of the dataset they use are very fine-grained and human-generated, making this approach feasible). They add a special filtering step, keeping only questions that can be answered with the caption but where two strong baselines fail to answer using the image (Qwen2.5-VL-7B and Qwen2.5-VL-32B).
For stages 2 and 3 (textual and visual reasoning), they use existing reasoning datasets that they filter.

They experiment with Qwen3-VL-8B-Instruct and Qwen2.5-VL7B-Instruct as base models and compare with a few open-weight reasoning VLMs trained on top of these same base models. They evaluate on a set of visual math reasoning and general visual perception benchmarks. They show the positive effect of this preliminary perception-oriented finetuning.

**Compliance With Llm Reviewing Policy:**

Affirmed.

**Final Justification:**

The authors have performed many extra experiments during the rebuttal, addressing my main concern (weakness of evaluation and domain gap). The performance gain is limited but consistent across models and benchmarks.
The final version of the paper will need substantial work to improve presentation and clarity, and would strongly benefit from experiments on larger-scale models (above 8B). Nevertheless, I think this paper should be accepted, and I increased my score.

**Key Questions For Authors:**

1. Reasoning data curation, "We retain samples that require both accurate perception and multi-step reasoning" --> how do you perform this filtering?

2. Figure 5: Does this thinking cost analysis lead to the same tendencies on perception vs. visual maths benchmark test sets, respectively?

3. Just making sure: for the merged training, the samples of the 3 stages are randomly shuffled?

4. What was the motivation for disabling the visual encoder for the merged training baseline but not the staged one?

**Limitations:**

Yes.

**Strengths And Weaknesses:**

**Soundness**:

Soundness of main result: The main experiment (Table 1) shows that fine-tuning on the created dataset consistently improves the base model performance.
- The fine-tuned VLM is also compared with a few baselines trained on top of the same base model. However, we don't know much about the baselines in Table 1, so it's hard to conclude why your model outperforms them. Do they all do RL training with math reasoning-only datasets, without any perception-oriented method (in terms of prompting, reward, training data, training strategy...)? Explaining these baselines in more detail would help interpret the results and draw conclusions beyond the fact that training on the created dataset is helpful.
- The evaluation is done on a large set of benchmarks, either math-domain reasoning-oriented or general-domain perception-only. Note that AOKVQA and POPE and pretty old benchmarks that are very easy for current SOTA models, even at the 7 and 8B scale. Hence, the margin of progression is very limited, which is visible especially in Figure 3, and would explain the less convincing results.

Soundness of ablations:
- A very large number of ablations are performed on every aspect of the curriculum training.
- The visual encoder is disabled (=frozen?) for the merged training baseline but not the staged one. Even if common practice is to disable it, it makes the comparison between the two baselines less fair, and it is not clear why this choice was made. Hence, it's hard to interpret the results of Table 2 comparing staged and merged training.

Reproducibility:
- A few missing pieces of information on the training data creation make it hard to interpret the results, and to reproduce the method; the filtering of the reasoning data is not explained at all, and the size of the datasets used for each stage is missing.
- Results in Table 1 for the baseline R1-OneVision are surprisingly low compared to the baseline, and quite different from the ones reported in the published paper (https://openaccess.thecvf.com/content/ICCV2025/papers/Yang_R1-Onevision_Advancing_Generalized_Multimodal_Reasoning_through_Cross-Modal_Formalization_ICCV_2025_paper.pdf). Same for the performance of the base model. Benchmark-wise, the largest gap is with WeMath (37.03 for the baseline on your paper and 61.0 for the linked one).


**Presentation**

Good writing overall.

The formalisation in 3.1.1 is making the reading more complicated than it should be, and doesn't seem to be used in the rest of the paper. You could remove equations 1 to 3.

Similarly, the GRPO training details (3.2.1) do not seem to differ from the original paper's formalisation. In that case, it's better to just refer to the paper, unless there are specific design choices that should be highlighted.

The conclusion is very repetitive.

A few missing pieces of information (see soundness), and limited discussion of the comparison with baselines.

Details:
- Bolding results in Table 1 separately for the two base models would be helpful to better interpret the results.
- Correct bolding in Table 3, AVG column for Qwen3.


**Significance and Originality**:

Visual reasoning is a very important and widely studied problem, and the focus on improving perception is very relevant and intensively studied in the community.
The capability-based curriculum learning method is well justified and seems to be effective.

There is a strong separation of domains between reasoning ability (stages 2 and 3 of training with math and scientific domain reasoning and images, evaluation on visual math benchmarks) and perception ability (general domain images for stage 1 training and for benchmarks).
First, it's unclear how that affects the transfer of ability. One could consider multi-step logical reasoning on general-domain images, and different levels of perception complexity on math domain images.
Second, this domain gap explains the results shown in Figure 3 and highlighted in the text (MMStar, POPE, A-OKVQA): fine-tuning a model on math reasoning only data, and then testing it on general-domain VQA, will for sure decrease its performance, and fine-tuning it on in-domain perception data will for sure increase it.

Overall, the method is quite simple (which is great) and the main conclusion is in line with the literature; but the amount of ablations and the new dataset creation are appreciated contributions.

---

> ### Author Rebuttal · Authors · 2026-03-31
>
> We thank the reviewer for recognizing the soundness of our results, ablation breadth, and dataset contribution. We address each concern below.
>
> ---
>
> ### W1: Baseline Descriptions in Table 1
>
> All baselines are open-weight reasoning VLMs with post-training (SFT/RL) for visual reasoning:
> - **GThinker-7B**: RL with cue-guided rethinking; includes perception via re-examining visual cues — most relevant comparator.
> - **MMR1-7B**: Variance-aware sampling + RL for math reasoning, no explicit perception optimization.
> - **OpenVLThinker-7B**: Iterative SFT-RL for vision-language reasoning, no dedicated perception stage.
> - **R1-OneVision-RL-7B**: Cross-modal RL for reasoning transfer from text to visual modalities.
> - **WeThink-7B**: GRPO on 120K+ multimodal QA with perception abilities embedded in reasoning pairs, but jointly trained.
> - **OneThinker-8B**: Two-stage SFT+RL on 600K corpus with EMA-GRPO, but no explicit perception-reasoning separation.
>
> **GThinker** and **WeThink** include perception components but integrate them into reasoning rather than training separately. Our method outperforms all, suggesting explicit staged decoupling is more effective. Details will be added in revision.
>
> ---
>
> ### W2: Replacing Saturated Benchmarks
>
> On **nine additional challenging benchmarks** (see full table in response to Reviewer bLqm, W1), staged training achieves **+4.03%** AVG over merged, with +9.13% (TextVQA), +6.99% (DynaMath), +6.36% (HallusionBench). This confirms advantages are clearer on harder benchmarks and the modest margins on A-OKVQA/POPE were due to ceiling effects. We will replace saturated benchmarks in revision.
>
> ---
>
> ### W3 & Q4: Visual Encoder Freezing Strategy
>
> The mixed strategy (frozen in Stage 2/text-only, unfrozen in Stages 1,3/visual) is a natural default. To address fairness, we ablated **all freezing strategies**:
>
> |Config|MathVista|MVision|WeMath|A-OKVQA|RWQA|POPE|AVG|
> |:----|:----:|:----:|:----:|:----:|:----:|:----:|:----:|
> |**Qwen2.5-VL-7B**||||||||
> |Staged/Mixed|**71.20**|21.71|38.29|**87.60**|**70.46**|**86.84**|**62.68**|
> |Staged/Freeze|70.00|20.72|38.38|86.90|69.02|86.33|61.89|
> |Staged/Open|70.50|**22.37**|**38.67**|87.16|69.93|86.08|62.45|
> |Merged/Freeze|70.90|20.39|36.67|85.59|70.07|84.35|61.33|
> |Merged/Open|70.00|20.39|37.43|86.03|69.28|84.74|61.31|
> |**Qwen3-VL-8B**||||||||
> |Staged/Mixed|**75.90**|28.62|56.10|86.29|74.51|**87.88**|68.22|
> |Staged/Freeze|75.20|30.26|**56.57**|86.11|**76.21**|87.42|**68.63**|
> |Staged/Open|75.30|**31.91**|54.67|**86.55**|74.90|87.18|68.42|
> |Merged/Freeze|74.30|25.99|54.38|85.33|73.86|87.40|66.88|
> |Merged/Open|73.80|28.95|55.43|85.50|75.56|87.19|67.74|
>
> Key findings: **(1)** Encoder freezing causes small differences within each paradigm (staged AVG: 61.89–62.68% on Qwen2.5-VL-7B). **(2)** Staged consistently outperforms merged under comparable configs (Freeze: 61.89 vs. 61.33, Open: 62.45 vs. 61.31 on Qwen2.5-VL-7B; Freeze: 68.63 vs. 66.88, Open: 68.42 vs. 67.74 on Qwen3-VL-8B). The advantage stems from the training paradigm, not encoder freezing.
>
> ---
>
> ### W4 & Q1: Details on Reasoning Data Curation
>
> - **Stage 1 (Perception):** ~3K QA pairs from DOCCI (15K images, human-annotated captions), after difficulty filtering (Section 3.1.1).
> - **Stage 2 (Text Reasoning):** ~13K text-only math samples from open-source datasets.
> - **Stage 3 (Visual Reasoning):** ~19K samples from VLAA-Thinking, keeping only instances correctly answered by Qwen2.5-VL-32B (6 samples), retaining visual math and ChartQA tasks [1].
>
> [1] Chou et al. Well-Formed, Ill-Grounded: Visual Alignment Gaps Where GPT-OSS Falls Short of Qwen, 2025.
>
> ---
>
> ### W5: R1-OneVision Result Discrepancy
>
> Discrepancies arise from evaluation protocol differences. We use a **unified framework** (VLMEvalKit) with consistent settings for all models, using official weights from HuggingFace. Gaps (e.g., WeMath 37.03% vs. 61.0%) likely reflect different benchmark versions/splits. All Table 1 models are evaluated under the **same protocol** for fair comparison. We will document all settings in revision.
>
> ---
>
> ### W6: Presentation Improvements
>
> We will: (1) simplify Section 3.1.1 formalization (remove Eqs. 1–3), (2) streamline GRPO in Section 3.2.1 by citing the original paper, (3) revise conclusion concisely, (4) bold Table 1 by base model groups, (5) correct Table 3 bolding, (6) add baseline/evaluation details.
>
> ---
>
> ### Q2: Thinking Cost Analysis
>
> Response lengths on visual math test sets (see table in response to Reviewer bprr, W1&Q1) show staged training produces **6.6%–12.6% shorter traces** across all benchmarks while achieving higher accuracy. This confirms correct perception reduces redundant re-checking, generalizing the trend in Figure 5 to held-out sets.
>
> ---
>
> ### Q3: Merged Training Data Shuffling
>
> Yes, all samples from three stages are **randomly shuffled** into one dataset for merged training, with identical hyperparameters and total training steps as staged training.

---

> > ### Author Rebuttal · Reviewer_jL8w · 2026-04-02
> >
> > Answer to Q1 is unclear, l.167 says "We retain samples that require both accurate perception and multi-step
> > reasoning, forming the dataset Dvis" but I don't see how keeping only instances correctly answered by Qwen2.5-VL-32B does that.
> >
> > The large amount of extra experiments during rebuttal is appreciated, and it makes the evaluation stronger. The proposed method has a clear, although small, impact on performance across benchmarks.

---

> > > ### Author Response · Authors · 2026-04-03
> > >
> > > Thank you for pointing this out. Our previous response was imprecise. Filtering with Qwen32B doesn't necessarily retain the samples that require "both accurate perception and multi-step reasoning”. Instead, the datasets we used (CLEVR-Math, GeoQA170K, MathPUMA, DocVQA, and ArxivQA) require perception and visual reasoning. When filtering with Qwen32B, we generate 8 rollouts per question, and only retain samples with pass rates falling in 1/8 \~ 7/8. This practice retains samples with moderate difficulty which could stabilize RL training [1].
> > >
> > > Thanks again for pointing out the fallacy, and we will revise the manuscript to make it precise.
> > >
> > > [1] Bae et al., 2025. Online Difficulty Filtering for Reasoning Oriented Reinforcement Learning.

---

### Official Review · Reviewer_GR7L · 2026-03-11

**Soundness:** 3
**Presentation:** 2
**Significance:** 3
**Originality:** 3
**Overall Recommendation:** 4
**Confidence:** 3

**Summary:**

The paper claims that in many cases model produces wrong answers not due to missing reasoning capabilities, but due to weak visual perception. The authors also claim that post-training based on the existing reasoning-biased datasets entangles visual perception, visual and textual reasoning. According to the authors, this can be solved through creation of **specialized data** targeting visual perception, textual and visual reasoning separately, **staged training** where the order of stage optimization is critical (visual perception preceding visual reasoning). Using **RLRV-based visual perception training** in contrast to SFT allows to achieve superior results.

**Compliance With Llm Reviewing Policy:**

Affirmed.

**Final Justification:**

The rebuttal addressed most of my concerns and demonstrated a positive attiture of the authors towards improving the paper. I believe that provided clarifications increased the readability and clarity of the paper and additional experiments made the method evaluation stronger. I would like to increase my score from 3 to 4.

**Key Questions For Authors:**

Q1. How does your approach separate visual perception errors in the generated answers from halucinations?
Q2. If visual perception and textual reasoning training stages can be exchanged, how can you make sure that the question is not answered exclusively based on the text caption and completely omits visual component?
Q3. What actually happens in 3-1-2, 2-3-1 and 1-3-2 orders? Do these experiments support the claim about the order importance between visual perception and reasoning as well?
Q4. How would you compare your work to other works that also separate visual perception from visual reasoning?

**Limitations:**

Limitations are not discussed in the paper explicitely.

**Strengths And Weaknesses:**

S1: Proposed method leads to a marginal improvement of Qwen2.5-VL-7B and Qwen3-VL-8B on several Math Reasoning and Perception benchmarks.
S2: The idea to separate visual perception and reasoning is meaningful and generally interesting

W1: The paper does not mention other works that propose similar idea of the importance of separation between visual perception and reasoning. However, there are several recent works, for example:
- Kamoi, Ryo, et al. "Visonlyqa: Large vision language models still struggle with visual perception of geometric information." arXiv preprint arXiv:2412.00947 (2024).
- Abdullaeva, Irina, et al. "NoReGeo: Non-Reasoning Geometry Benchmark." arXiv preprint arXiv:2601.10254 (2026)

W2: All experiment are performed on a similar scale Qwen 2.5 and 3 models. To make a conclusion about general usefullness of staged training and proposed data more experiments with other VLM families will be required

W3. Table 3 has an issue with highlighted highest results which leads to confusion and makes claims made in the introduction partially misleading.
Highlighting issue in:
- Qwen2.5-VL-7B model 2->1->3 order for Mvista, Mverse(VI) and WeMath Benchmarks has higher results.
- Qwen3-VL-8B in Perception task (AVG))
Claims in the paper text:
Intro (lines 104-107): Moreover, the order of stage optimization is critical, as visual perception serves as the fundamental scaffold that should be solidified before refining visual reasoning.
In my opinion, this should be reformulated.
Moreover, results in table 6 demonstrate that only on 2 out of 6 benchmarks the results of having separate visual perception and reasoning post-training in comparison to just visual reasoning lead to significant improvement.

---

> ### Author Rebuttal · Authors · 2026-03-31
>
> We thank the reviewer for recognizing the significance of separating visual perception from reasoning. We address each concern below.
>
> ---
>
> ### W1 & Q4: Related Works on Separating Perception from Reasoning
>
> We compare VisOnlyQA (Kamoi et al., 2024) and NoReGeo (Abdullaeva et al., 2026) with our work. **VisOnlyQA** benchmarks VLMs' geometric perception via vision-only questions, revealing models struggle with basic visual perception. **NoReGeo** isolates perception from reasoning in geometry, further exposing perception gaps. These share our motivation that perception is a bottleneck masked by emphasis on reasoning. Our contribution diverges in the **solution**: we address the deficit through a **training methodology** — a capability-based staged paradigm consolidating perception before reasoning, complementary to these diagnostic benchmarks.
>
> We incorporated VisOnlyQA in our evaluation: staged training achieves **64.00%** vs. merged 61.56% (**+2.44%**), directly addressing the perception gap VisOnlyQA identifies. We will discuss these works in revision.
>
> ---
>
> ### W2: Generalization on Other VLM Architectures
>
> We conducted experiments on the **InternVL** family (InternVL3.5-8B, InternVL3-8B) across ten benchmarks:
>
> |Model|MathVista|MathVision|MathVerse(VO)|MathVerse(VI)|WeMath|DynaMath|RealWorldQA|HallusionBench|BLINK|VisOnlyQA|AVG|
> |:----|:----:|:----:|:----:|:----:|:----:|:----:|:----:|:----:|:----:|:----:|:----:|
> |InternVL3.5-8B(Merged)|69.40|28.95|32.11|32.61|48.38|62.34|65.88|48.55|56.13|50.89|49.52|
> |InternVL3.5-8B(Staged)|**70.30**|**31.91**|**33.76**|**34.26**|**49.90**|**62.83**|**67.84**|**53.47**|**57.71**|**52.00**|**51.40**|
> |InternVL3-8B(Merged)|60.70|26.32|25.25|30.46|25.05|45.79|64.44|30.35|**48.45**|40.22|39.70|
> |InternVL3-8B(Staged)|**65.40**|**26.64**|**29.70**|**32.99**|**34.95**|**52.42**|**64.71**|**36.42**|46.87|**46.22**|**43.63**|
>
> Staged **consistently outperforms merged**: InternVL3.5-8B +**1.88%** AVG (HallusionBench +4.92%, MathVision +2.96%), InternVL3-8B +**3.93%** AVG (WeMath +9.90%, DynaMath +6.63%). This confirms generalization beyond the Qwen family. Results will be included in revision.
>
> ---
>
> ### W3 & Q3: Stage Order and Corrected Highlighting
>
> We will correct highlighted results. Our claim: **perception (Stage 1) should precede visual reasoning (Stage 3)**. Textual reasoning (Stage 2) is independent:
>
> - **1→2→3 ≈ 2→1→3** (both perception-first): Qwen2.5-VL-7B 42.26% vs. 42.91%; Qwen3-VL-8B 51.10% vs. 50.75%.
> - **3→2→1 degrades**: 37.70% on Qwen2.5-VL-7B (-4.56% vs. 1→2→3).
>
> We further compare perception→reasoning vs. reasoning-only on nine benchmarks (see table in response to Reviewer bprr, W1&Q1). Adding perception training improves AVG by **+2.30%** (DynaMath +12.28%). Combined with Appendix Table 6 (Stage 1→3: 68.27% vs. Stage 3: 67.33%), this consistently shows perception provides a foundation for reasoning. We will reformulate claims and correct highlights.
>
> ---
>
> ### Q1: Separating Visual Perception Errors from Hallucinations
>
> Following "More Thinking, Less Seeing?" framework, perception errors and hallucinations both refer to **incorrect understanding of image details/spatial relationships**. We use Claude-Haiku-4.5 as evaluator detecting: (1) object identification errors, (2) attribute errors (color, size, shape), (3) spatial relation errors, (4) counting errors, (5) text reading errors. Perception errors subsume visual hallucinations. Our finding (86.9% of errors from perception) captures the full spectrum of visual misunderstanding. The full evaluation prompt can be found in Figure 8 of our original paper.
>
> ---
>
> ### Q2: Ensuring Questions Are Not Answered Solely from Text Captions
>
> We appreciate the question and would like to clarify a potential misunderstanding. The text captions are used **only during the data synthesis pipeline** (to generate perception-focused questions via an LLM) and are **never provided to the model during training or evaluation**. Specifically:
>
> 1. We collect image-caption pairs from the DOCCI dataset, where captions are fine-grained and human-annotated.
> 2. We provide the captions to an LLM (Qwen2.5-72B) to generate perception-focused QA pairs.
> 3. During GRPO training (Stage 1), the model receives only the **image and question** as input — the caption is not included. The model must generate the correct answer by **fully understanding the visual content** of the image, as there is no textual shortcut available.
>
> Furthermore, our perception difficulty filtering step (Section 3.1.1) explicitly removes questions that the base VLMs can already answer correctly from the image, ensuring that the retained questions target genuine perception gaps. The experimental results in the table in response to Reviewer bprr, W1&Q1, confirm that such perception training effectively improves the model's visual grounding capability, leading to better visual reasoning performance downstream. We will make this distinction clearer in the revised manuscript.

---

> > ### Author Rebuttal · Reviewer_GR7L · 2026-04-03
> >
> > Thanks for the detailed rebuttal and additional experiment results provided!
> > The question regarding the benefits of proposed method for larger scale models remains, as currently all the tested models of a max size 8B.
> > Overall, we appreciate expected revisions and additional experiment results and believe they make the paper more clear.

---

> > > ### Author Response · Authors · 2026-04-04
> > >
> > > We sincerely thank the reviewer for acknowledging our rebuttal and for the positive recognition of our additional experiments and expected revisions.
> > >
> > > Regarding the follow-up question on larger-scale models: we would like to respectfully note that the original review (W2) stated that "*more experiments with other VLM families will be required*," without specifically requesting experiments at larger model scales. In direct response to this concern, we conducted extensive new experiments on two InternVL model series (InternVL3-8B and InternVL3.5-8B), demonstrating consistent improvements of +3.93% and +1.88% on average, respectively.
> > >
> > > We fully agree that validating our approach at larger scales is an important direction. However, scaling to significantly larger models (e.g., 32B or 72B) requires substantial GPU resources that are beyond the scope of the current rebuttal period. We would also like to note that the 7–8B scale is the standard and widely adopted setting for research papers in the VLM post-training literature — all of our baselines (GThinker, MMR1, OpenVLThinker, R1-OneVision, WeThink, OneThinker) are evaluated at this scale as well. We are committed to extending our validation to larger models in future work and will clearly state this in the revised manuscript.
> > >
> > > We believe that the **consistent improvement across four distinct model series** — Qwen2.5-VL-7B, Qwen3-VL-8B, InternVL3-8B, and InternVL3.5-8B — spanning two different VLM families with different architectures, provides strong evidence for the effectiveness and generalizability of our staged training paradigm. We hope that our rebuttal has sufficiently addressed the concerns raised in the original review, and we kindly ask the reviewer to consider these additional results when finalizing the assessment.

---

### Official Review · Reviewer_bprr · 2026-03-12

**Soundness:** 2
**Presentation:** 3
**Significance:** 2
**Originality:** 2
**Overall Recommendation:** 4
**Confidence:** 3

**Summary:**

This paper studies why vision language models fail on visual reasoning tasks. The authors argue that the main problem is not reasoning but visual perception. They show that once the model makes a perception error, longer reasoning usually cannot fix it. To address this, they separate the training process into three stages: visual perception, textual reasoning, and visual reasoning. Each stage uses different training data. They also show that visual perception is better improved with reinforcement learning than caption based supervised fine tuning. Experiments on several benchmarks show that staged training improves both perception and reasoning performance. The model also produces shorter reasoning traces and achieves better results on visual math and perception tasks compared with the base models.

**Compliance With Llm Reviewing Policy:**

Affirmed.

**Final Justification:**

The three run results make the proof for the approach's effectiveness somehow more solid. Therefore, I decided to raise my score from 3->4.

**Key Questions For Authors:**

Why is it necessary to separate perception and reasoning into different stages? In real model responses, these abilities often appear together. Can you add an experiment to justify your approach?


The improvement over merged training seems small in many cases. How significant are these gains in practice?


The effect of stage order seems limited. Some different orders give similar results. Does this mean the training order is not very critical?

**Limitations:**

The performance improvement is relatively small. In many benchmarks the gain over merged training is only a few points. It is unclear whether this improvement is meaningful.


The novelty of the method is limited. The idea of staged or curriculum style training has been explored in many previous works.


The paper claims that staged training is clearly better than merged training, but the experimental gap is not very large.


The approach mainly studies two backbone models. It is not clear if the conclusion generalizes to other VLM architectures.

**Strengths And Weaknesses:**

**Strengths**:

The motivation is reasonable and intuitive. The paper argues that perception errors often cause reasoning failures in VLMs. This idea is easy to understand and makes sense.

The experiments are fairly thorough. The authors test the method on several benchmarks and multiple model backbones. They also include ablations such as merged training vs staged training and different stage orders.

**Weaknesses**:

I am not fully convinced by the need to separate these abilities so strictly. In real model responses, perception and reasoning often happen together. A model may first describe the image and then reason about it in the same answer. It is unclear why these abilities must be learned in clearly separated stages. A good reasoning model should not treat these skills as isolated.


There are several overclaims in the paper. The improvement of staged training over merged training is actually small in many cases. It is usually only a few points. The influence of the stage order is also limited. Some different orders produce very similar results. This makes the claim that staged training is significantly better less convincing.

---

> ### Author Rebuttal · Authors · 2026-03-31
>
> We thank the reviewer for recognizing the intuitive motivation and thoroughness of our experiments. We address each concern below.
>
> ---
>
> ### W1 & Q1: Why Separate Perception and Reasoning into Different Stages?
>
> Though perception and reasoning co-occur at inference, whether they should be **learned together** is underexplored, which is our central question. Our hypothesis: perception is a **prerequisite** for visual reasoning. Unreliable perception causes reasoning on incorrect premises, leading to repeated re-checking and lengthy traces (Figures 1, 4).
>
> We compare perception→reasoning (Stage 1→3) vs. reasoning-only (Stage 3) on **nine benchmarks**:
>
> |Model|MathVerse(VO)|TextVQA|HallusionBench|BLINK|ChartQA|CV-Bench|DynaMath|VisOnlyQA|V\*Bench|AVG|
> |:----|:----:|:----:|:----:|:----:|:----:|:----:|:----:|:----:|:----:|:----:|
> |Qwen3-VL-8B(Perc→Reas)|**38.83**|**73.99**|**54.62**|62.91|**59.96**|**80.21**|**65.65**|**60.22**|**83.77**|**64.46**|
> |Qwen3-VL-8B(Reas-only)|37.06|73.31|52.89|**62.97**|57.84|79.29|53.37|60.00|82.72|62.16|
>
> Perception training improves AVG by **+2.30%** (DynaMath +12.28%). Appendix Table 6 corroborates: Stage 1→3 achieves 68.27% vs. Stage 3 at 67.33%.
>
> The staged model also produces **shorter reasoning traces**:
>
> |Model|MathVista|MathVision|MathVerse(VO)|WeMath|
> |:----|:----:|:----:|:----:|:----:|
> |Qwen3-VL-8B(Staged)|1325.89|2930.41|1541.89|1745.69|
> |Qwen3-VL-8B(Merged)|1420.30|3163.41|1764.93|1906.07|
>
> Staged achieves **higher accuracy with 6.6%–12.6% shorter traces**, as correct perception eliminates redundant re-checking — strong evidence that separating perception addresses a fundamental bottleneck.
>
> ---
>
> ### W2, Q2, L1, & L3: Significance of Improvement
>
> On original saturated benchmarks (A-OKVQA, POPE), margins appeared modest. On **nine challenging benchmarks** (see table in response to Reviewer bLqm, W1), staged achieves **+4.03%** AVG, with +9.13% (TextVQA), +6.99% (DynaMath), +6.36% (HallusionBench), outperforming merged on all nine. Cross-architecture gains are consistent: InternVL3.5-8B +1.88%, InternVL3-8B +3.93% (see response to Reviewer GR7L, W2). A +4% gain from reordering the same data without extra compute is meaningful and cost-free.
>
> ---
>
> ### W3 & Q3: Effect of Stage Order
>
> Our claim: **perception should precede visual reasoning**. Textual reasoning is independent. Table 3 shows: 1→2→3 ≈ 2→1→3 (Qwen2.5-VL-7B: 42.26% vs. 42.91%; Qwen3-VL-8B: 51.10% vs. 50.75%), while 3→2→1 (visual-reasoning before the perception) degrades to 37.70% (-4.56%). Our ablation above (W1&Q1) confirms perception→reasoning gives +2.30% over reasoning-only. The similarity of 1→2→3 and 2→1→3 confirms the perception-before-reasoning principle drives the gain. We will refine claims accordingly.
>
> ---
>
> ### L2: Novelty — Comparison with Existing Curriculum Learning
>
> Existing VLM curricula (Curr-ReFT, PC-GRPO) organize by **difficulty** (easy→hard). We propose a **capability-based** dimension (perception→reasoning). These are **orthogonal and complementary**. We compare four settings on Qwen3-VL-8B: (1) **None**: merge all training data and random shuffling; (2) **Capability**: our staged training (perception→textual reasoning→visual reasoning); (3) **Difficulty**: merged data ordered by descending pass rate (K=8 rollouts on base model), learning easy samples first; (4) **Cap+Diff**: staged training with difficulty curriculum within each stage.
>
> |Curriculum|MVision|MathVerse(VO)|WeMath|DynaMath|RWQA|CV-Bench|V\*Bench|AVG|
> |:----|:----:|:----:|:----:|:----:|:----:|:----:|:----:|:----:|
> |None(Merged)|28.95|36.68|55.43|54.15|75.56|78.53|80.63|58.56|
> |Capability|28.62|39.97|56.10|61.14|74.51|79.62|83.77|60.53|
> |Difficulty|24.67|40.86|54.10|67.07|72.68|79.36|83.77|60.36|
> |Cap+Diff|**33.22**|**41.75**|**57.43**|**67.49**|**75.82**|**80.95**|**84.29**|**62.99**|
>
> Both curriculum dimensions independently improve over none-curriculum (Capability: 60.53%, Difficulty: 60.36%, vs. None: 58.56%). Combining them yields the best **62.99%** AVG (+4.43% over none), with notable gains on DynaMath (+13.34%) and MathVision (+4.27%). This demonstrates our capability-based curriculum is a genuinely new, complementary contribution.
>
> ---
>
> ### L4: Generalization on Other Architectures
>
> We extended to **InternVL** (full table in response to Reviewer GR7L, W2). Staged training improves InternVL3.5-8B by **+1.88%** and InternVL3-8B by **+3.93%** across ten benchmarks, confirming cross-architecture generalization.

---

> > ### Author Rebuttal · Reviewer_bprr · 2026-04-03
> >
> > Thanks for the detailed rebuttal. I agree with your claim "slightly gain from reordering the same data without extra compute is meaningful and cost-free", however, I strongly suggest the author add an experiment to run for three times for a single benchmark to eliminate the randomness of performance in VLMs. And the slightly gain (e.g., CV-Bench gains for like less than 1% in W1 & Q1) seems only to be differ from one question, which cannot fully confirm the effectiveness of your approach.

---

> > > ### Author Response · Authors · 2026-04-06
> > >
> > > We sincerely thank the reviewer for the continued engagement and for the constructive suggestion regarding statistical robustness. We have evaluated both staged and merged training on Qwen3-VL-8B and Qwen2.5-VL-7B across **all 15 benchmarks for three independent runs** and report the averaged results:
> > >
> > > | Model | MathVista | MathVision | MathVerse (VO) | MathVerse (VI) | WeMath | DynaMath | A-OKVQA | RealWorldQA | MMStar | HallusionBench | POPE | BLINK | CV-Bench | VisOnlyQA | V\*Bench | AVG |
> > > | :---- | :----: | :----: | :----: | :----: | :----: | :----: | :----: | :----: | :----: | :----: | :----: | :----: | :----: | :----: | :----: | :----: |
> > > | Qwen3-VL-8B (Staged) | **76.20** | **28.84** | **40.01** | **43.23** | **56.86** | **68.67** | **86.99** | **74.08** | **73.15** | **55.88** | **87.63** | **65.40** | **80.70** | **64.08** | **85.51** | **65.82** |
> > > | Qwen3-VL-8B (Merged) | 72.93 | 26.86 | 35.91 | 38.88 | 52.98 | 66.58 | 85.42 | 73.77 | 70.20 | 50.97 | 87.18 | 62.97 | 78.31 | 62.15 | 80.28 | 63.03 |
> > > | Qwen2.5-VL-7B (Staged) | **71.50** | **20.94** | **32.95** | **38.28** | 37.02 | **54.05** | **87.07** | 69.54 | **64.82** | **39.98** | **87.06** | **55.74** | **75.95** | **48.82** | **76.79** | **57.37** |
> > > | Qwen2.5-VL-7B (Merged) | 69.63 | 19.52 | 29.40 | 34.01 | **37.21** | 52.84 | 85.85 | **70.15** | 63.31 | 36.42 | 84.67 | 54.57 | 74.72 | 47.85 | 76.62 | 55.78 |
> > >
> > > With three-run averaging, the improvements remain **consistent and robust**:
> > >
> > > - **Qwen3-VL-8B:** Staged training outperforms merged training by **+2.79%** on average (65.82% vs. 63.03%), with staged training winning on **all 15 benchmarks**. The gains are particularly substantial on MathVerse-VO (+4.10%), MathVerse-VI (+4.35%), DynaMath (+2.09%), MMStar (+2.95%), and HallusionBench (+4.91%) — well beyond single-question margins.
> > > - **Qwen2.5-VL-7B:** Staged training outperforms merged training by **+1.59%** on average (57.37% vs. 55.78%), with staged training winning on **13 out of 15 benchmarks**. Notable gains include MathVerse-VO (+3.55%), MathVerse-VI (+4.27%), HallusionBench (+3.56%), and POPE (+2.39%).
> > >
> > > We acknowledge the reviewer's observation that some individual benchmark differences (e.g., CV-Bench) are small. However, the three-run averaged results demonstrate that the overall pattern is reliable and not attributable to random fluctuation: staged training achieves broad, consistent gains across the majority of benchmarks, with the most pronounced improvements on challenging tasks such as MathVerse, DynaMath, MMStar, and HallusionBench.
> > >
> > > Combined with the consistent improvements observed across **four model series** (Qwen2.5-VL-7B, Qwen3-VL-8B, InternVL3-8B, InternVL3.5-8B), we believe these results firmly establish that our capability-based staged curriculum is an effective and robust training paradigm. As a new curriculum dimension, it offers a **cost-free** improvement (same data, same total compute, different ordering) that is **complementary** to the existing difficulty-based curriculum. We hope these additional results address the reviewer's concern regarding statistical robustness, and we kindly ask the reviewer to consider them when finalizing the assessment.

---

### Official Review · Reviewer_bLqm · 2026-03-13

**Soundness:** 2
**Presentation:** 2
**Significance:** 2
**Originality:** 2
**Overall Recommendation:** 4
**Confidence:** 4

**Summary:**

This paper proposes to decouple VLM post-training into three stages: visual perception, text reasoning, and visual reasoning. First, RLVR is used to enhance perception, and then reasoning is trained. It is shown that staged training is better than combined training, which improves accuracy and shortens the reasoning length.

**Compliance With Llm Reviewing Policy:**

Affirmed.

**Key Questions For Authors:**

Supplement the generalization verification with other architectures such as LLaVA and InternVL. Further verify the effectiveness of this paradigm.

**Limitations:**

Look at the weaknesses section.

**Strengths And Weaknesses:**

**Strengths**

(1) Experiments demonstrate that perceptual-first + RLVR training outperforms merged training, improving accuracy and shortening inference length.

**Weaknesses**

(1) Table 1 shows that three-stage training did not achieve state-of-the-art (SOTA) on multiple datasets. Furthermore, key perceptual benchmarks such as BLINK and V* are lacking.

(2) Table 3 shows that different stage orders each have their advantages, but the optimal order is not clearly defined.

(3) Experiments are based only on the Qwen series models, lacking generalization validation with other architectures such as LLaVA and InternVL.

---

> ### Author Rebuttal · Authors · 2026-03-30
>
> We sincerely thank the reviewer for the thoughtful evaluation and for recognizing the effectiveness of our perception-first staged training paradigm, including its ability to improve accuracy and shorten inference length over merged training. We address each concern below.
>
> ---
>
> ### W1: Evaluation on Additional Perceptual and Visual Math Benchmarks
>
> We appreciate this suggestion and conducted additional experiments on nine new benchmarks covering visual reasoning (MathVerse-Vision-Only, ChartQA, DynaMath) and perception (TextVQA, HallusionBench, BLINK, CV-Bench, VisOnlyQA, V\*Bench). Notably, these include **BLINK and V*Bench**, which the reviewer specifically suggested. Results comparing staged and merged training on Qwen3-VL-8B are shown below:
>
> |Model|MathVerse(VO)|TextVQA|HallusionBench|BLINK|ChartQA|CV-Bench|DynaMath|VisOnlyQA|V*Bench|AVG|
> |:----|:----:|:----:|:----:|:----:|:----:|:----:|:----:|:----:|:----:|:----:|
> |Qwen3-VL-8B (Staged)|**39.97**|**74.59**|**59.54**|**64.12**|**62.56**|**79.62**|**61.14**|**64.00**|**83.77**|**65.48**|
> |Qwen3-VL-8B (Merged)|36.68|65.46|53.18|61.34|61.56|78.53|54.15|61.56|80.63|61.45|
>
> Staged training outperforms merged training on all nine benchmarks, with an average gain of +4.03%. Gains are especially large on TextVQA (+9.13%), HallusionBench (+6.36%), V*Bench (+3.14%), and DynaMath (+6.99%). These results confirm that staged training yields broad, consistent improvements across diverse settings, including the key perceptual benchmarks previously missing from our evaluation. We will incorporate these results in the revised manuscript.
>
> ---
>
> ### W2: Clarification on the Optimal Stage Order
>
> We thank the reviewer for raising this point. We clarify that visual perception (Stage 1) is a prerequisite for visual reasoning (Stage 3), so Stage 1 should be consolidated before Stage 3. By contrast, textual reasoning (Stage 2) is relatively independent of visual perception, and the order between Stages 1 and 2 has limited impact.
>
> This principle is consistently supported by our results. As shown in Table 3, 1→2→3 and 2→1→3 yield comparable performance (e.g., on Qwen2.5-VL-7B, 42.3% vs. 42.9% visual math AVG; on Qwen3-VL-8B, 51.10% vs. 50.75%), since both place perception before visual reasoning. In contrast, 3→2→1 clearly degrades performance on Qwen2.5-VL-7B (37.70% visual math AVG), because visual reasoning is trained before perception is strengthened.
>
> To further validate this principle, we provide additional experiments comparing perception→reasoning (Stage 1→3) with reasoning-only (Stage 3) on nine new benchmarks:
>
> |Model|MathVerse(VO)|TextVQA|HallusionBench|BLINK|ChartQA|CV-Bench|DynaMath|VisOnlyQA|V\*Bench|AVG|
> |:----|:----:|:----:|:----:|:----:|:----:|:----:|:----:|:----:|:----:|:----:|
> |Qwen3-VL-8B(Perc→Reas)|**38.83**|**73.99**|**54.62**|62.91|**59.96**|**80.21**|**65.65**|**60.22**|**83.77**|**64.46**|
> |Qwen3-VL-8B(Reas-only)|37.06|73.31|52.89|**62.97**|57.84|79.29|53.37|60.00|82.72|62.16|
>
> Adding perception training before visual reasoning improves average accuracy by +2.30%, with the largest gains on DynaMath (+12.28%) and ChartQA (+2.12%). These results confirm that perception serves as a scaffold for visual reasoning, and placing it earlier in the pipeline yields consistently better outcomes. We will refine the discussion and correct the bolding in Table 3 in the revised manuscript.
>
> ---
>
> ### W3 & Q1: Lacking Generalization Validation on Other VLM Architectures
>
> Thank you for suggesting validation on other model families. We conducted experiments on the **InternVL** family (InternVL3.5-8B, InternVL3-8B) with staged and merged training across ten benchmarks (see full table in our response to Reviewer GR7L, W2).
>
> Staged training **consistently outperforms merged**: InternVL3.5-8B improves by **+1.88%** (51.40% vs. 49.52%), with gains on HallusionBench (+4.92%) and MathVision (+2.96%). InternVL3-8B improves by **+3.93%** (43.63% vs. 39.70%), with notable gains on WeMath (+9.90%) and DynaMath (+6.63%). These cross-architecture results confirm our approach generalizes beyond the Qwen family. We will include these in revision.

---

> > ### Author Rebuttal · Reviewer_bLqm · 2026-04-03
> >
> > Many thanks for the detailed rebuttal and the substantial additional experiments. I also carefully reviewed the other reviewers' comments and the corresponding responses.
> >
> > However, I notice that the newly added tables only compare staged training versus merged training, without including the base model baselines (i.e., Qwen3-VL-8B (Base), Qwen2.5-VL-7B (Base), InternVL3.5-8B (Base), and InternVL3-8B (Base)) across all evaluated datasets. These baselines are essential for assessing the absolute improvement brought by the proposed method over the pretrained models, rather than only the relative gain between two training strategies. I would encourage the authors to include base model performance in all tables so that readers can fully contextualize the effectiveness of the proposed approach.

---

> > > ### Author Response · Authors · 2026-04-06
> > >
> > > We sincerely thank the reviewer for the continued engagement and for the constructive suggestion to include base model baselines. We have evaluated all four base models across the benchmarks and present the complete results below:
> > >
> > > | Model | MathVista | MathVerse (VO) | MathVerse (VI) | WeMath | DynaMath | MMStar | HallusionBench | BLINK | VisOnlyQA | V\*Bench | AVG |
> > > | :---- | :---- | :---- | :---- | :---- | :---- | :---- | :---- | :---- | :---- | :---- | :---- |
> > > | **InternVL3.5-8B (Base)** | 60.70 | 11.93 | 5.08 | 40.10 | 44.85 | 43.27 | 39.88 | 46.66 | 36.89 | 43.98 | 37.33 |
> > > | **InternVL3.5-8B (Merged)** | 69.40 | 32.11 | 32.61 | 48.38 | 62.34 | **64.40** | 48.55 | 56.13 | 50.89 | **62.83** | 52.76 |
> > > | **InternVL3.5-8B (Staged)** | **70.30** | **33.76** | **34.26** | **49.90** | **62.83** | 61.60 | **53.47** | **57.71** | **52.00** | 61.26 | **53.71** |
> > > | **InternVL3-8B (Base)** | 18.10 | 17.13 | 22.34 | 3.14 | 13.15 | 46.33 | 30.06 | 48.55 | 41.56 | 56.54 | 29.69 |
> > > | **InternVL3-8B (Merged)** | 60.70 | 25.25 | 30.46 | 25.05 | 45.79 | 51.87 | 30.35 | 48.45 | 40.22 | **61.26** | 41.94 |
> > > | **InternVL3-8B (Staged)** | **65.40** | **29.70** | **32.99** | **34.95** | **52.42** | **53.00** | **36.42** | **46.87** | **46.22** | 59.16 | **45.71** |
> > > | **Qwen2.5-VL-7B (Base)** | 68.40 | 24.11 | 25.00 | 30.86 | 51.54 | 63.67 | **40.46** | 55.39 | **49.56** | **76.96** | 48.60 |
> > > | **Qwen2.5-VL-7B (Merged)** | 69.75 | 29.57 | 34.23 | **37.24** | 52.89 | 63.33 | 36.56 | 54.60 | 47.95 | 76.83 | 50.30 |
> > > | **Qwen2.5-VL-7B (Staged)** | **71.45** | **32.93** | **38.13** | 36.88 | **54.03** | **64.79** | 39.95 | **55.71** | 48.67 | 76.70 | **51.92** |
> > > | **Qwen3-VL-8B (Base)** | 72.40 | 26.90 | 31.09 | 50.86 | **66.43** | 70.00 | 32.95 | **68.39** | 61.78 | 83.77 | 56.46 |
> > > | **Qwen3-VL-8B (Merged)** | 73.80 | 36.68 | 40.36 | 55.43 | 54.15 | 70.60 | 53.18 | 61.34 | 61.56 | 80.63 | 58.77 |
> > > | **Qwen3-VL-8B (Staged)** | **75.90** | **39.97** | **43.78** | **56.10** | 61.14 | **73.07** | **59.54** | 64.12 | **64.00** | **83.77** | **62.14** |
> > >
> > > Across all four model series, staged training consistently achieves the highest performance, demonstrating both substantial absolute improvements over base models and meaningful relative gains over merged training. We will include base model results in all tables in the revised manuscript to provide this complete picture. We hope these comprehensive results sufficiently address the reviewer's concern.

---

### Decision · Program_Chairs · 2026-04-30

**Decision:**

Accept (regular)

**Comment:**

This paper studies the effects of perception and reasoning in VLM post-training. The authors propose decomposing VLM post-training into three stages: visual perception, textual reasoning, and visual reasoning. Each stage requires different training strategies and data. Experimental results show promising performance, validating the proposed framework.

All reviewers have a positive final judgment on this submission. At the initial review stage, reviewers raised concerns about the experimental results, mostly regarding the inclusion of additional models and datasets, reporting average scores over multiple runs, and providing additional ablations. The authors properly addressed and responded to these comments. All reviewers are satisfied with the newly provided evidence. Other than the experimental results, there were no other major issues raised by the reviewers. As a result, I recommend an accept for this submission.